# TOWARDS ADVERSARIALLY ROBUST CLIP: A HIERARCHICAL MODEL FUSION METHOD USING OPTIMAL TRANSPORT

## ABSTRACT

In recent years, multimodal models such as CLIP have achieved impressive performance but remain vulnerable to adversarial perturbations. Although adversarial training can enhance robustness, it often leads to overfitting toward specific attack types. One solution for improving generalization is to integrate multiple diverse and adversarially trained submodels, but this strategy could incur high test-time cost. To achieve a promising tradeoff between robust generalization and efficiency, we consider to design an optimal transport (OT) based model fusion method, which is called "HOT-CLIP (Hierarchical Optimal Transport Fusion for CLIP)". Although several OT based model fusion methods have been proposed before, they cannot be easily adapted to solve our problem, since they may suffer the issues like parameter misalignment when dealing with highly diverse and multimodal submodels. Our proposed method constructs diverse submodels by varying both attack methods and textual prompts, and integrates them via a hierarchical two-level OT fusion method. The intra-attack fusion first aligns and merges models within the same attack family, and the inter-attack fusion subsequently combines these aligned models across different attacks. Through this carefully crafted fusion strategy, HOT-CLIP can significantly improve the accuracy for alignment and reduce the total occupied memory. More importantly, the obtained robust visual encoder can be deployed without additional inference-time cost. In our experiments, the results on multiple vision-language tasks demonstrate that HOT-CLIP can greatly enhance the model's adversarial robustness while maintaining competitive clean accuracy.

## 1 INTRODUCTION

The rapid advancement of large vision–language models (LVLMs) (Zhang et al., 2024) has significantly reshaped the landscape of artificial intelligence. By jointly learning from visual and textual modalities, LVLMs demonstrate strong generalization ability across a wide range of downstream tasks, including image classification (Radford et al., 2021), image retrieval (Li et al., 2022), image captioning (Hu et al., 2022), and multimodal reasoning (Yin et al., 2024). A key step in this progress is the development of **alignment** models (Radford et al., 2021; Li et al., 2022). Among them, Contrastive Language–Image Pretraining (CLIP) (Radford et al., 2021) is a representative framework that leverages large-scale contrastive pretraining on image–text pairs to align visual and textual representations effectively.

Although CLIP demonstrates remarkable performance across a wide range of large vision–language tasks, it still faces significant challenges in robustness and reliability. In particular, CLIP's vision encoder is vulnerable to adversarial perturbations: even small, carefully crafted changes to the input image can induce substantial misalignment between visual and textual representations, ultimately leading to errors in downstream tasks. Prior work suggests that this vulnerability may be partly attributed to the high dimensionality and local linearity of deep visual feature spaces (Goodfellow et al., 2015). This vulnerability is especially concerning in safety-critical domains such as medical imaging (Javed et al., 2024) and autonomous driving (Rossolini et al., 2023), where incorrect predictions may cause severe consequences. Figure 1 provides an example to illustrate the adversarial impact on image captioning performance.

To address these vulnerabilities, various defense strategies have been explored. "Test-time" defenses attempt to mitigate adversarial effects without modifying the training process. One recently proposed approach is CIDER (Xu et al., 2024), which detects adversarial images by measuring the semantic distance between the original and denoised inputs. Another example, SmoothVLM (Sun et al., 2024) introduces controlled noise to mitigate the effects of localized adversarial perturbations. However, since test-time defenses do not modify model parameters, they are often limited in their ability to address the underlying vulnerabilities of CLIP's vision encoder.

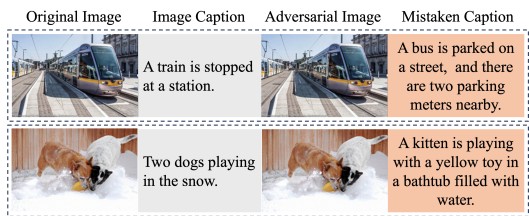

Figure 1: The *adversarial images* are generated using the PGD attack under the $\ell_\infty$ norm with $\epsilon = 2/255$, and the *mistaken captions* are obtained by applying LLaVA to these perturbed inputs.

A more effective strategy is *adversarial training*, where models are explicitly optimized on adversarial examples to improve robustness (Madry et al., 2018; Schlarmann et al., 2024). For example, RobustCLIP (Schlarmann et al., 2024) incorporates adversarial perturbations during training to maintain alignment, while Sim-CLIP (Hossain & Imteaj, 2024b) enforces representation consistency between clean and perturbed samples. However, adversarial training is often prone to overfitting to specific attack patterns and exhibits limited generalization to unseen perturbations (Rice et al., 2020). To overcome the limited generalization of adversarial training, ensemble-based methods (Dong et al., 2020; Hu et al., 2024; Zhang et al., 2025a) have been explored to improve robustness by combining multiple models, which can collectively handle a wider variety of perturbations. However, conventional ensemble methods are challenging to deploy on LVLMs, as these models already impose substantial computational demands (Zhang et al., 2025b), and ensembling will further introduce significant additional memory and inference overhead. Beyond ensembling, **model fusion** techniques (Smith & Gashler, 2017) aim to integrate the parameters of multiple networks rather than merely combining their outputs. The major difference between ensembling and fusion is that ensembling scales inference cost with the number of submodels, while fusion only produces a single model.

Nevertheless, the challenge of standard fusion methods (e.g., weight averaging) (Smith & Gashler, 2017) is the lack of one-to-one correspondence between the parameters from different models. Namely, for two different models, the neurons respectively locating in the same position of them may not be functionally similar (a simple example is given in Figure 2 of Section 2). Thus, it is necessary to align the neurons before parameter averaging. **Optimal Transport (OT)** (Peyré & Cuturi, 2019), as formally defined in Section 2, provides a principled metric that quantifies the distance between two probability distributions by computing the minimal cost of transporting one distribution to match the other. In the context of model fusion, OT can be used to compute a transport matrix $T$ that aligns neurons across models before performing averaging-based fusion (Singh & Jaggi, 2020; Imfeld et al., 2024). Although OT can partially mitigate the parameter misalignment issue, its effectiveness might be constrained when applied to highly diverse models. This is because standard OT establishes correspondences based on geometric distances in parameter space (e.g., Euclidean or cosine), which may not really capture the semantic consistency (Chuang et al., 2023). Consequently, parameters that are close under such geometric measures can still encode distinct underlying features, particularly when the models are trained under different conditions (e.g., different data distributions or architectures).

In summary, applying OT Fusion to adversarially trained CLIP submodels is not straightforward, due to a challenging dilemma. **On the one hand, to ensure model's robustness, the submodels need to be as diverse as possible,** since submodels trained under different adversarial attacks develop distinct mappings in the input space, resulting in diverse representations. This diversity produces complementary decision boundaries across submodels, collectively enhancing the final fused model's robustness to a wider range of adversarial perturbations. **On the other hand, to ensure the accuracy of OT Fusion, the submodels need to be as similar as possible.** As explained above, OT primarily aligns parameters based on geometric metrics rather than semantic consistency, which may reduce its effectiveness when the submodels are highly diverse.

**Our contributions.** To achieve a pleasant trade-off between the above two aspects, we propose Hierarchical Optimal Transport Fusion method for CLIP (HOT-CLIP), a novel framework that con-

structs diverse adversarial submodels and hierarchically fuses them via optimal transport to produce a single robust CLIP visual encoder. To the best of our knowledge, this is also the first work to investigate model fusion problem in the context of adversarial robustness.

– HOT-CLIP adopts a carefully crafted two-stage procedure, which first constructs diverse adversarial submodels, and then hierarchically fuses them into a single robust visual encoder. Stage 1 constructs a set of diverse submodels by varying both the adversarial attack settings used during training (training data) and the textual prompts (training labels). This diversity ensures that the resulting submodels capture complementary robustness patterns. Stage 2 hierarchically fuses these submodels via optimal transport, effectively balancing the trade-off between submodel diversity and alignment. It first aligns and fuses models within the same attack family (intra-attack), consolidating prompt-induced diversity while ensuring that the models are sufficiently similar for effective OT alignment. It then fuses the resulting models across different attacks (inter-attack), integrating attack-based diversity to produce the final robust model.

–Then, we conduct extensive experiments on three representative tasks for vision-language models, image classification, visual question answering (VQA), and image captioning. Across these tasks, HOT-CLIP consistently enhances adversarial robustness over existing methods, with relative robust score improvements of around $2.6\%$ for image classification, around $20.4\%$ for VQA, and around $16.5\%$ for image captioning. At the same time, its clean performance remains highly competitive. These results demonstrate that HOT-CLIP can effectively navigate the robustness–accuracy trade-off, and has the potential to establish a new state-of-the-art for adversarially robust multimodal models.

## 2 PRELIMINARIES

Due to space constraint, an extended review of related work is presented in the Appendix B. In this section, we briefly introduce the preliminaries related to our study, including the structure of CLIP (Radford et al., 2021), adversarial training (Madry et al., 2018), and optimal transport fusion (Peyré & Cuturi, 2019; Singh & Jaggi, 2020).

Let $\mathcal{X}$ denote the image space and $\mathcal{T}$ denote the text space. CLIP consists of two modality-specific encoders: an image encoder $f_{\theta_{\text{img}}} : \mathcal{X} \to \mathbb{R}^d$ and a text encoder $f_{\theta_{\text{txt}}} : \mathcal{T} \to \mathbb{R}^d$, where $\theta$ denotes the model parameters. In particular, we denote the parameters of the image and text encoders as $\theta_{\text{img}}$ and $\theta_{\text{txt}}$, respectively. These encoders map images $x \in \mathcal{X}$ and text descriptions $t \in \mathcal{T}$ into a shared $d$-dimensional embedding space.

**Definition 2.1** (**CLIP Classifier**). *Consider a $K$-class classification task. Let $\mathcal{C} = \{c_1, c_2, \ldots, c_K\}$ denote the set of candidate classes, and $\mathcal{Y} = \{1, 2, \ldots, K\}$ denote the corresponding label set. For each class $c_k$, define a textual prompt $t_k$ associated with class $c_k$ (e.g., $t_k =$ "a photo of $c_k$"). The CLIP classifier $g : \mathcal{X} \to \mathbb{R}^K$ is defined as*

$$g(x)_k = \cos\left(f_{\theta_{img}}(x), f_{\theta_{txt}}(t_k)\right), \quad k = 1, \ldots, K, \tag{1}$$

*where $f_{\theta_{img}}$ and $f_{\theta_{txt}}$ denote the image and text encoders, respectively, $x \in \mathcal{X}$ is the input image and $\cos(\cdot, \cdot)$ computes the cosine similarity between normalized embeddings.*

**Definition 2.2** (**Adversarial Example**). *Given a classifier $g : \mathcal{X} \to \mathbb{R}^K$ and a clean input image $x \in \mathcal{X}$ with true label $y \in \mathcal{Y}$, an adversarial example is a perturbed input*

$$x' = x + \eta, \ \|\eta\|_p \leq \epsilon, \ s.t. \ argmax \ g(x') \neq y, \tag{2}$$

*where $\eta$ is a small perturbation constrained within an $L_p$ -ball of radius $\epsilon$, such that the classifier misclassifies the perturbed input (i.e. $g(x') \neq y$). To obtain such perturbations, adversarial attacks can be categorized into two types:*

*(i) Untargeted attack: the adversary aims to maximize the loss for the true label:*

$$\eta^* = \arg \max_{\|\eta\|_p \leq \epsilon} \mathcal{L}(g(x + \eta), y), \tag{3}$$

*where $\mathcal{L}$ is the loss function (e.g. cross-entropy).*

*(ii) Targeted attack: the adversary seeks to misclassify the input as a specific target class $y_{target} \neq y$:*

$$\eta^* = \arg \min_{\|\eta\|_p \leq \epsilon} \mathcal{L}(g(x + \eta), y_{target}). \tag{4}$$

To defend against such adversarial perturbations, adversarial training (Madry et al., 2018) introduces these adversarial examples into the learning process. In our setup, only the parameters of the image encoder are updated, while the text encoder remains frozen. Specifically, the training objective is formulated as the following min–max optimization:

$$\min_{\theta_{img}} \mathbb{E}_{(x,y)\sim\mathcal{D}} \left[ \max_{\|\eta\|_p \leq \epsilon} \mathcal{L}\big(g(x+\eta; \theta_{img}, \theta_{text}), y\big) \right]. \tag{5}$$

where $\mathcal{D}$ is the dataset. The inner maximization identifies the most challenging adversarial perturbations within the allowed $\ell_p$ norm bound, while the outer minimization updates the image encoder to correctly classify these perturbed inputs, thereby enhancing adversarial robustness.

**Definition 2.3** (Optimal Transport Distance (Peyré & Cuturi, 2019))**.** *Let $\mu = \sum_{i=1}^{n} \alpha_i \, \delta(a^{(i)})$ and $\nu = \sum_{j=1}^{m} \beta_j \, \delta(b^{(j)})$ be two empirical probability measures, where $a^{(i)} \in \mathcal{P}$ and $b^{(j)} \in \mathcal{Q}$ are support points, and $\delta(\cdot)$ denotes the Dirac measure assigning unit mass. Here, $\mathcal{P}$ and $\mathcal{Q}$ represent the spaces of source and target points (e.g., neuron embeddings to be aligned). We define a transport cost function $C : \mathcal{P} \times \mathcal{Q} \to \mathbb{R}^+$, which quantifies the cost of transporting unit mass from $a^{(i)}$ to $b^{(j)}$. The optimal transport distance between $\mu$ and $\nu$ is defined as*

$$\mathrm{OT}(\mu,\nu) = \min_{T \in \Pi(\mu,\nu)} \mathbb{E}_{(a,b)\sim T}\big[\, C(a,b) \,\big], \tag{6}$$

*where $\Pi(\mu,\nu)$ denotes the set of couplings with marginals $\mu$ and $\nu$.*

In the above definition, the minimizer $T^* \in \Pi(\mu,\nu)$, called the *optimal transport plan*, defines a minimal-cost correspondence between the support points of $\mu$ and $\nu$. In the context of model fusion, $T^*$ can be used to align neurons or parameter vectors across models, providing a principled way to combine them while minimizing misalignment. Based on optimal transport, OT Fusion (Singh & Jaggi, 2020) aligns two (or more) neural networks in a layer-wise manner. In each layer, neurons are treated as points ($a$ and $b$), with their associated weights or activations serving as feature representations. Assuming a uniform probability measure over neurons ($\mu$ and $\nu$), the OT problem is solved between corresponding layers to obtain the transport matrix. The transport matrix is then used to align the current layer, and

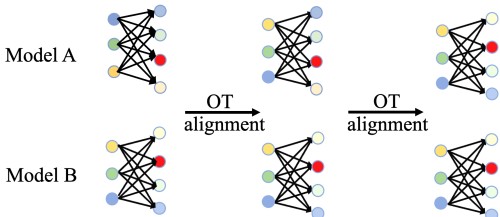

Figure 2: Neuron alignment via OT. Assume the neurons with same color are functionally similar. The original models "A" and "B" exhibit a permutation in neuron correspondence (left figure); for example, the first neuron in the first layer of A is blue, but the neuron in the same position of B is yellow. In the middle figure, we align the first layer via OT; then, the second layer is aligned in the right figure.

the aligned weights are subsequently averaged to produce the fused layer. Applying this procedure sequentially across layers yields a coherent fusion of the models. In Figure 2, we provide a simple two-layer example, and the full details of OT Fusion are shown in Appendix C.1.

## 3 METHODOLOGY

In this section, we first present the high-level idea of our method in Section 3.1, and then detail the technical components, including the construction of diverse submodels and the hierarchical OT Fusion in Sections 3.2 and 3.3, respectively.

### 3.1 OVERVIEW OF OUR METHOD

Our goal is to enhance the robustness of CLIP's visual encoder against adversarial perturbations. The key idea is to leverage diversity among adversarially trained submodels and integrate them through OT Fusion. Figure 3 illustrates the overall framework.

**Stage 1: Diverse Adversarial Submodels Construction.** According to the adversarial training objective introduced in Definition 2.2, the parameters of a model are influenced by multiple factors, such as the training data, model architecture, and optimization algorithm. Among these, training

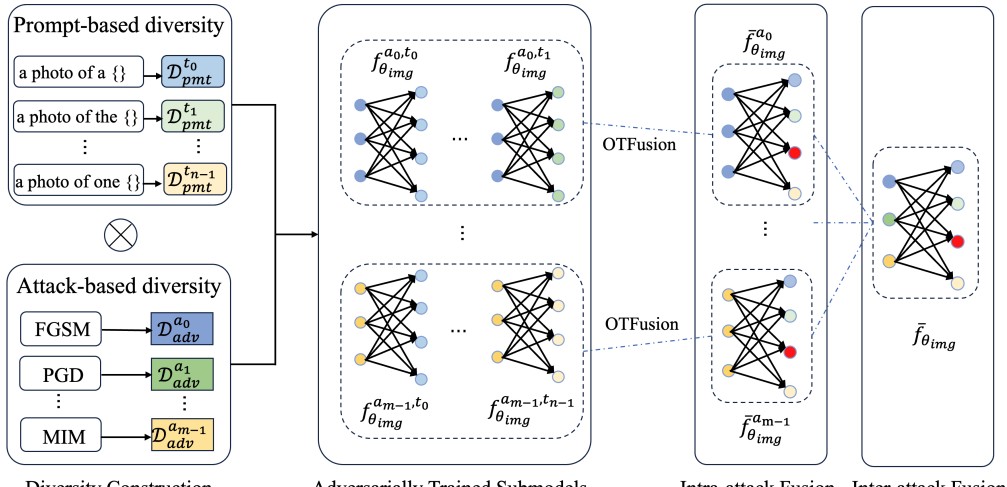

Figure 3: Overview of HOT-CLIP. Diverse submodels are first constructed along two axes: prompt-based diversity ($n$ different textual templates) and attack-based diversity ($m$ different adversarial attacks). The hierarchical OT Fusion method is then applied to integrate these $n \times m$ submodels. In the Intra-attack OT Fusion stage, submodels within the same attack family but with different prompts are aligned and fused via optimal transport. In the Inter-attack OT Fusion stage, the first-level fused models from different attack families are further integrated, yielding the final robust visual encoder.

data plays a particularly crucial role, as variations in data directly affect the learned decision boundaries and representations. In our study, all submodels share the same architecture and optimization algorithm, and the **diversity** arises solely from the differences between those specifically "modified" data distributions. Concretely, we train the submodels on adversarial examples generated by different attack algorithms, so that the desired robustness can be diversified across multiple perturbation types. Additionally, a single text template is often insufficient to fully capture the alignment between images and textual labels; for example, the original CLIP (Radford et al., 2021) used about 80 templates. To further enhance the diversity, we vary both the adversarial attack settings and the textual prompts, so that the submodels are trained under different distributions of adversarial examples and supervision.

**Stage 2: Hierarchical OT Fusion.** Directly fusing these diverse adversarially trained CLIP submodels may be suboptimal. Theoretically, the parameters that are close under geometric measures can encode distinct underlying features, particularly for diversely trained submodels (Chuang et al., 2023; Nguyen et al., 2023; Ormaniec et al., 2025). Even using OT (as illustrated in Figure 2), the highly diverse submodels could still yield non-ignorable error for the final fusion. Moreover, simultaneously fusing all submodels can take a rather large amount of memory space, as a great number of parameters need to be stored and aligned within the same period during the fusion. To neatly circumvent "direct" OT Fusion, we propose a hierarchical two-level OT Fusion framework. The key idea is to control both the similarity and the number of submodels involved at each fusion step, thereby improving alignment quality while keeping memory usage manageable. At the first level, submodels trained under the same adversarial attack but with different textual prompts are grouped and fused via OT. This ensures high internal homogeneity within each group and limits the number of models fused simultaneously. At the second level, the first-level fused models, already aligned in the label space, are further integrated across different attacks. By progressively fusing more homogeneous submodels, this hierarchical design can effectively preserve parameter alignment, leverages complementary robustness, and maintains an affordable memory size.

## 3.2 CONSTRUCTION OF DIVERSE ADVERSARIAL SUBMODELS

We construct a set of diverse submodels by varying both the attack settings used during adversarial training and the textual prompts used for image–text alignment.

**Attack-based diversity.** As discussed in Section 3.1, adversarial training typically improves robustness against the specific perturbations seen during training but generalizes poorly to unseen attacks. To mitigate this limitation, we construct submodels under different adversarial settings. Formally,

let $\mathcal{A}$ denote a set of adversarial attack methods. Given a clean dataset $\mathcal{D}$ and an attack method $a \in \mathcal{A}$, we define the attack-specific dataset as

$$\mathcal{D}_{\text{adv}}^{(a)} = \{(x + \eta_a(x), y) \mid (x, y) \in \mathcal{D}\} \tag{7}$$

where $\eta_a$ denotes the perturbation generated under method $a$ (e.g., FGSM (Goodfellow et al., 2015), PGD (Madry et al., 2018), MIM (Dong et al., 2018)) with its own radius $\epsilon$ and norm constraint. Training a visual encoder on $\mathcal{D}_{\text{adv}}^{(a)}$ yields a submodel specialized to adversarial perturbations of $a$.

**Prompt-based diversity.** Beyond attack-based diversity, CLIP-style models also rely on text–image alignment, which introduces another source of diversity. We consider multiple textual prompt sets $\mathcal{T}$. For a given prompt set $t \in \mathcal{T}$, the prompt-specific dataset is defined as

$$\mathcal{D}_{\text{prompt}}^{(t)} = \{(x, y^{(t)}) \mid (x, y) \in \mathcal{D}\}, \tag{8}$$

where $y^{(t)}$ is the textual representation for class $y$ under prompt $t$. This generates the submodels with different label alignment characteristics in the embedding space.

For each combination of attack $a \in \mathcal{A}$ and prompt $t \in \mathcal{T}$, we define a fully diversified dataset that integrates $\mathcal{D}_{\text{adv}}^{(a)}$ and $\mathcal{D}_{\text{prompt}}^{(t)}$:

$$\mathcal{D}_{\text{div}}^{(a,t)} = \{(x + \eta_a(x), y^{(t)}) \mid (x, y) \in \mathcal{D}\}. \tag{9}$$

Then, we train a submodel $f_{\theta_{\text{img}}}^{(a,t)}$ on $\mathcal{D}_{\text{div}}^{(a,t)}$, where only the parameters $\theta_{\text{img}}$ of the image encoder are updated and the text encoder $\theta_{\text{text}}$ remains fixed. The resulting family of submodels is $\mathcal{M} = \{f_{\theta_{\text{img}}}^{(a,t)} \mid a \in \mathcal{A}, t \in \mathcal{T}\}$.

## 3.3 HIERARCHICAL OT FUSION

We then introduce the two-level fusion procedure, consisting of the L-level (Language-level) fusion and the V-level (Visual-level) fusion. At the L-level, submodels trained under the same adversarial attack but using different textual prompts are fused. This step consolidates the diversity arising from multiple prompts. At the V-level, the resulting L-level fused models are further fused across different adversarial attacks, integrating the diversity introduced by varying attack methods. Due to the space limit, we leave the full HOT-CLIP algorithm to Appendix C.

**L-Level: Intra-attack Fusion.** For a fixed attack $a \in \mathcal{A}$, the set of submodels $\mathcal{M}_a = \{f_{\theta_{\text{img}}}^{(a,t)} \mid t \in \mathcal{T}\}$ share similar robustness properties but differ in feature alignment due to different textual prompts. The L-level fused model $\bar{f}_{\theta_{\text{img}}}^{(a)}$ is obtained by averaging the weights of submodels, after aligning them to an "anchor" model (which can be any arbitrary submodel selected from $\mathcal{M}_a$). Formally, for each layer $\ell$, the fused weight is

$$W_\ell^{(a)} = \frac{1}{|\mathcal{T}|} \sum_{t \in \mathcal{T}} W_\ell^{(a,t),\text{aligned}}, \tag{10}$$

where $W_\ell^{(a,t),\text{aligned}}$ denotes the aligned weights of submodel $f_{\theta_{\text{img}}}^{(a,t)}$. To obtain the aligned weights $W_\ell^{(a,t),\text{aligned}}$, we first align the columns of the current layer's weight $W_\ell^{(a,t)}$ using the transport matrix from the previous layer, $T_{\ell-1}^{(a,t)}$. These partially aligned weights are then used to compute the transport matrix $T_\ell^{(a,t)}$ for the current layer, which is subsequently applied to align the rows of $W_\ell^{(a,t)}$, yielding the fully aligned weights $W_\ell^{(a,t),\text{aligned}}$. Formally, $W_\ell^{(a,t),\text{aligned}}$ is calculated by

$$W_\ell^{(a,t),\text{aligned}} = T_\ell^{(a,t)\top} W_\ell^{(a,t)} T_{\ell-1}^{(a,t)}, \tag{11}$$

The right multiplication by $T_{\ell-1}^{(a,t)}$ aligns the columns of $W_\ell^{(a,t)}$ to the anchor model, while the left multiplication by $T_\ell^{(a,t)\top}$ aligns the rows with respect to the anchor model. For more details, please see Appendix C.1.

**V-Level: Inter-attack Fusion.** After intra-attack fusion, we obtain a set of fused models corresponding to each attack method, denoted as $\{\bar{f}_{\theta_{\text{img}}}^{(a)} \mid a \in \mathcal{A}\}$. We then apply OT Fusion to these models. Specifically, we compute the OT matrix $T^{(a)}$ between the layers of each submodel and an arbitrarily selected anchor from $\{\bar{f}_{\theta_{\text{img}}}^{(a)} \mid a \in \mathcal{A}\}$. For a given layer $\ell$, the aligned weights are obtained as

$$W_\ell^{(a),\text{aligned}} = T_\ell^{(a)\top} W_\ell^{(a)} T_{\ell-1}^{(a)}, \tag{12}$$

where $W_\ell^{(a)}$ denotes the weight matrix of layer $\ell$ in $\bar{f}_{\theta_{\text{img}}}^{(a)}$, $T_{\ell-1}^{(a)}$ is the transport map from the previous layer, and $T_\ell^{(a)\top}$ is the transpose of the current layer's transport map. Similar to the L-level fusion, the right multiplication by $T_{\ell-1}^{(a)}$ aligns the columns of $W_\ell^{(a)}$ with the anchor model, while the left multiplication by $T_\ell^{(a)\top}$ aligns the rows. The aligned weights are then averaged across all attack-specific models:

$$W_\ell^{\text{fused}} = \frac{1}{|\mathcal{A}|} \sum_{a \in \mathcal{A}} W_\ell^{(a),\text{aligned}}. \tag{13}$$

Repeating this procedure for all layers yields the final robust model $\bar{f}_{\theta_{\text{img}}}$, which integrates both prompt-level and attack-level diversity.

**Memory usage during the fusion.** By controlling the diversity at each step, this hierarchical approach could ensure that only relatively similar submodels are aligned at a time. Specifically, the peak memory size occupied by the submodels is lowered from $\mathcal{O}(|\mathcal{A}||\mathcal{T}| \cdot U)$ for naive global fusion to $\mathcal{O}(\max\{|\mathcal{A}|, |\mathcal{T}|\} \cdot U)$, where $U$ denotes the memory usage of a single submodel; the detailed analysis is provided in Appendix C.3.

To illustrate some theoretical intuition for why the hierarchical fusion behaves reasonably, we include the Lemma 3.1, which provides an upper bound on the distance between the hierarchical fused center and the global Wasserstein barycenter (Agueh & Carlier, 2011). This bound suggests that the hierarchical fused center remains controlled by the average distance of the individual submodels to the global barycenter. We add the proof in Appendix C.2.

**Lemma 3.1.** *Let $\mu_{a,t}$ denote the submodels trained under attack type $a$ and prompt variant $t$. Let $\mu_a^*$ be the intra-attack OT barycenter, $\mu_{\text{hier}}^*$ the hierarchical barycenter obtained from $\{\mu_a^*\}$, and $\mu_{\text{global}}^*$ the global Wasserstein barycenter computed over all submodels. Let $W_c$ denote the Wasserstein distance. Then*

$$W_c(\mu_{\text{hier}}^*, \mu_{\text{global}}^*) \leq \frac{4}{|\mathcal{A}\mathcal{T}|} \sum_{a,t} W_c(\mu_{a,t}, \mu_{\text{global}}^*).$$

## 4 EXPERIMENTS

In this section, we evaluate the adversarial robustness of CLIP's visual encoder enhanced with our hierarchical OT Fusion (HOT-CLIP) framework. We conduct experiments on three representative multimodal tasks (zero-shot image classification (Radford et al., 2021), visual question answering (VQA) (Goyal et al., 2017), and image captioning (Hu et al., 2022)) under diverse adversarial attack scenarios.

### 4.1 EXPERIMENTAL SETUP

**Implementation Details.** We adopt a two-stage training pipeline that combines the construction of diverse adversarial submodels and hierarchical OT fusion, using CLIP (Radford et al., 2021) with ViT-L/14 visual encoders as the backbone. First, we construct diverse adversarial submodels by training CLIP for two epochs on ImageNet, where each submodel is adversarially trained under a specific attack method (e.g., FGSM (Goodfellow et al., 2015), PGD (Madry et al., 2018), MIM (Dong et al., 2018)) and with a textual prompt randomly sampled from the 80 templates introduced in (Radford et al., 2021). Next, we perform the hierarchical OT fusion. Within each attack family, submodels trained with different textual prompts are fused via OT, where layer-wise weight matrices are represented as neuron embeddings and the Sinkhorn algorithm (Cuturi, 2013) is used to compute the transport plan. The resulting fused model is fine-tuned for one epoch under the corresponding attack setting with a standard text template ("a photo of a . . . "). Finally, the first-level

Table 1: **Clean and adversarial evaluation on image classification datasets of CLIP model.** Models are trained on ImageNet, all other datasets are zero-shot. Robustness is assessed using AutoAttack with the $l_\infty$ norm and perturbation bound $\epsilon = 2/255$ and $\epsilon = 4/255$ .The last column shows the average accuracy across datasets. Bold indicates the best performance in each column.

| Eval. | Vision encoder | Zero-shot datasets | | | | | | | | | | | | | | Avg. |
|---|---|---|---|---|---|---|---|---|---|---|---|---|---|---|---|---|
| | | ImageNet | CalTech | Cars | CIFAR10 | CIFAR100 | DTD | EuroSAT | FGVC | Flowers | ImageNet-R | ImageNet-S | PCAM | OxfordPets | STL-10 | |
| clean | CLIP | 74.9 | 82.6 | 78.6 | 95.6 | 72.7 | 55.7 | 63.5 | 33.3 | 79.8 | 87.7 | 58.6 | 52.2 | 92.1 | 99.6 | **73.3** |
| | TeCoA | 77.4 | 77.3 | 35.3 | 80.6 | 51.0 | 39.5 | 24.0 | 13.2 | 40.5 | 73.1 | 54.5 | 49.8 | 77.3 | 93.9 | 56.2 |
| | FARE | 69.2 | 84.0 | 63.1 | 76.7 | 57.2 | 44.0 | 20.2 | 23.3 | 57.1 | 80.9 | 57.2 | 50.2 | 87.5 | 96.7 | 61.9 |
| | **HOT-CLIP** | 69.9 | 85.1 | 64.0 | 78.7 | 60.0 | 46.2 | 18.0 | 21.8 | 56.9 | 80.8 | 59.9 | 49.6 | 87.1 | 96.3 | 62.5 |
| $\epsilon = 2/255$ | CLIP | 0.0 | 0.0 | 0.0 | 0.0 | 0.0 | 0.0 | 0.0 | 0.0 | 0.0 | 0.0 | 0.1 | 0.0 | 0.0 | 0.0 | 0.0 |
| | TeCoA | 62.5 | 70.0 | 17.5 | 60.9 | 34.0 | 27.2 | 14.4 | 5.7 | 24.1 | 58.8 | 44.0 | 47.2 | 68.3 | 87.4 | 44.4 |
| | FARE | 52.1 | 76.8 | 29.8 | 56.5 | 36.2 | 28.4 | 12.2 | 8.0 | 28.3 | 61.0 | 41.9 | 50.2 | 71.5 | 89.7 | 45.9 |
| | **HOT-CLIP** | 53.4 | 79.5 | 31.8 | 58.3 | 37.2 | 31.8 | 12.4 | 8.1 | 29.7 | 60.9 | 44.6 | 49.6 | 71.2 | 91.0 | **47.1** |
| $\epsilon = 4/255$ | CLIP | 0.0 | 0.0 | 0.0 | 0.0 | 0.0 | 0.0 | 0.0 | 0.0 | 0.0 | 0.0 | 0.0 | 0.0 | 0.0 | 0.0 | 0.0 |
| | TeCoA | 48.2 | 61.4 | 8.7 | 37.3 | 20.2 | 17.6 | 11.6 | 2.3 | 12.5 | 41.5 | 34.5 | 38.1 | 55.7 | 74.6 | 33.1 |
| | FARE | 33.0 | 64.6 | 12.5 | 34.5 | 20.5 | 17.0 | 11.2 | 2.0 | 12.3 | 40.4 | 31.3 | 50.2 | 50.5 | 74.6 | 32.4 |
| | **HOT-CLIP** | 34.7 | 66.5 | 15.8 | 38.1 | 20.9 | 19.6 | 10.3 | 2.9 | 12.5 | 39.4 | 32.8 | 49.6 | 51.8 | 77.0 | **33.7** |

fused models from different attacks are integrated through OT Fusion, followed by an additional one epoch of unsupervised adversarial fine-tuning. Further details for hyperparameters and implementation are deferred to Appendix C.1.

**Baselines.** We compare our method against recent state-of-the-art approaches that aim to improve the adversarial robustness of vision–language models: TeCoA (Mao et al., 2023), which provides a systematic analysis of adversarial robustness in CLIP-like models and proposes tailored fine-tuning strategies to improve zero-shot performance under adversarial settings. FARE (Schlarmann et al., 2024), which introduces adversarial perturbations directly into the visual embedding space and fine-tunes the image encoder in an unsupervised manner.

### 4.2 EVALUATION ON PERFORMANCE

**Zero-shot Image Classification.** We evaluate clean and robust accuracies of the CLIP models on ImageNet and 13 zero-shot datasets mentioned in Appendix D. For each dataset, class names are combined with a predefined set of prompt templates. Zero-shot classification is then performed as described in Definition 2.1. To evaluate the adversarial robustness of the models, we adopt AutoAttack (Croce & Hein, 2020) under the $\ell_\infty$ norm with perturbation radii of $\epsilon = 2/255$ and $\epsilon = 4/255$, each run for 100 iterations. As shown in Table 1, HOT-CLIP achieves the second-best clean accuracy among all methods, slightly lower than the original CLIP. Under adversarial perturbations, it achieves a relative improvement in robustness of 2.6% at $\epsilon = 2/255$ and 1.8% at $\epsilon = 4/255$. These results indicate that our method improves adversarial robustness while maintaining competitive clean accuracy on image classification.

**Image Captioning.** We further evaluate our method on image captioning, where the model generates natural language descriptions of images. We report the results using the CIDEr score (Vedantam et al., 2015), a widely adopted metric for measuring the quality of generated captions. The experiments are conducted on COCO (Lin et al., 2014) and Flickr30k (Young et al., 2014), using two representative LVLMs: OpenFlamingo-9B (OF) (Alayrac et al., 2022) and LLaVA-1.5-7B (Liu et al., 2023). For clean evaluation, we use the full validation sets; for adversarial evaluation, we randomly sample 500 images from each dataset, using a similar evaluation setup as Schlarmann & Hein (2023). The adversarial robustness is tested with AutoAttack (Croce & Hein, 2020) under the $\ell_\infty$ norm with $\epsilon \in \{2/255, 4/255\}$, using 100 iterations. As shown in Table 2, HOT-CLIP consistently improves robustness across both datasets. For LLaVA-7B, HOT-CLIP achieves a relative

Table 2: **Evaluation of LVLMs using different CLIP encoders on image captioning**. Results are reported for OpenFlamingo and LLaVA on two image captioning datasets, measured using CIDEr. The last column shows the average CIDEr score across datasets. Bold indicates the best performance in each column.

| VLM | Vision encoder | COCO | | | Flickr30k | | | Average over datasets | | |
|---|---|---|---|---|---|---|---|---|---|---|
| | | clean | $\epsilon = \frac{2}{255}$ | $\epsilon = \frac{4}{255}$ | clean | $\epsilon = \frac{2}{255}$ | $\epsilon = \frac{4}{255}$ | clean | $\epsilon = \frac{2}{255}$ | $\epsilon = \frac{4}{255}$ |
| LLaVA-7B | CLIP | 122.2 | 3.2 | 2.4 | 79.1 | 1.4 | 0.9 | **100.6** | 2.3 | 1.65 |
| | TeCoA | 93.9 | 40.8 | 16.9 | 50.9 | 26.3 | 16.5 | 72.4 | 33.5 | 16.7 |
| | FARE | 105.8 | 50.1 | 33.2 | 64.7 | 28.5 | 20.1 | 85.2 | 39.3 | 26.6 |
| | **HOT-CLIP** | 110.4 | 56.5 | 35.5 | 74.6 | 43.1 | 26.5 | 92.5 | **49.8** | **31.0** |
| OF-9B | CLIP | 85.2 | 1.6 | 1.3 | 63.8 | 0.6 | 0.5 | **74.5** | 1.1 | 0.9 |
| | TeCoA | 73.5 | 31.6 | 21.2 | 43.5 | 10.4 | 10.2 | 58.5 | 21.0 | 15.7 |
| | FARE | 81.5 | 33.2 | 22.8 | 54.6 | 16.1 | 10.5 | 68.0 | 24.6 | 16.6 |
| | **HOT-CLIP** | 87.9 | 36.7 | 26.0 | 55.2 | 19.1 | 11.7 | 71.5 | **27.9** | **18.8** |

Table 3: **Evaluation of LVLMs using different CLIP encoders on VQA**. Results are reported on VQAv2 and TextVQA, measured by accuracy. The last column shows the average accuracy across datasets. Bold indicates the best performance in each column.

| VLM | Vision encoder | TextVQA | | | VQAv2 | | | Average over datasets | | |
|---|---|---|---|---|---|---|---|---|---|---|
| | | clean | $\epsilon = \frac{2}{255}$ | $\epsilon = \frac{4}{255}$ | clean | $\epsilon = \frac{2}{255}$ | $\epsilon = \frac{4}{255}$ | clean | $\epsilon = \frac{2}{255}$ | $\epsilon = \frac{4}{255}$ |
| LLaVA-7B | CLIP | 37.8 | 0.2 | 0.0 | 72.4 | 2.6 | 0.2 | **55.1** | 1.4 | 0.1 |
| | TeCoA | 19.4 | 12.8 | 8.9 | 63.4 | 40.4 | 29.5 | 41.4 | 26.6 | 19.2 |
| | FARE | 27.5 | 15.4 | 9.1 | 65.6 | 40.9 | 29.7 | 46.5 | 28.1 | 19.4 |
| | **HOT-CLIP** | 25.3 | 17.7 | 12.8 | 68.8 | 43.8 | 34.6 | 47.0 | **30.7** | **23.7** |
| OF-9B | CLIP | 21.0 | 0.0 | 0.0 | 46.2 | 3.7 | 0.5 | 33.6 | 1.9 | 0.2 |
| | TeCoA | 12.4 | 2.9 | 1.8 | 45.6 | 25.5 | 22.3 | 29.0 | 14.2 | 12.1 |
| | FARE | 17.0 | 3.5 | 2.6 | 43.2 | 24.0 | 20.7 | 30.1 | 13.7 | 11.6 |
| | **HOT-CLIP** | 22.5 | 5.2 | 5.0 | 51.5 | 29.1 | 24.0 | **37.0** | **17.1** | **14.5** |

improvement of 26.7% at $\epsilon = 2/255$ and 16.5% at $\epsilon = 4/255$; for OF-9B, the corresponding relative gains are 13.4% and 13.3%. These results demonstrate that HOT-CLIP effectively enhances adversarial robustness in image captioning while preserving competitive clean performance.

**VQA.** We also evaluate our method on the task of Visual Question Answering, where the model is required to provide accurate answers to natural language questions based on image inputs. For evaluation, we consider two widely used VQA benchmarks: VQAv2 (Goyal et al., 2017) and TextVQA (Singh et al., 2019). Adversarial evaluation uses the same model and attack settings as in the image captioning experiments, i.e., OpenFlamingo-9B and LLaVA-1.5-7B, with AutoAttack under the $\ell_\infty$ norm ($\epsilon = 2/255$ and $4/255$, 100 iterations). Table 3 reports the VQA accuracy under clean and adversarial settings, showing that HOT-CLIP consistently enhances robustness across both benchmarks while maintaining competitive clean performance. For LLaVA-7B, HOT-CLIP achieves a relative improvement of 9.2% at $\epsilon = 2/255$ and 22.2% at $\epsilon = 4/255$; for OF-9B, the corresponding relative gains are 20.4% and 19.8%.

**Summary of other experimental results.** Due to space constraint, additional ablation studies and extended evaluations are provided in Appendix E. We conduct ablation studies to examine the effects of backbone choice, submodel pool composition, adversarial training strength, and fusion strategies. We further evaluate our method on additional tasks, including targeted attacks and hallucination phenomena in large vision-language models. The results demonstrate that our framework consistently enhances adversarial robustness across tasks and configurations, while remaining effective across different model architectures.

Table 4: Comparison of training cost, fusion efficiency, and memory usage.

| Method | Training Time (GPU hours) | Fusion Time (h) | Fusion Memory (GB) |
|---|---|---|---|
| FARE | 196 | - | - |
| Direct OT Fusion | 1288 | 1.5 | 641 |
| HOT-CLIP | 1552 | 1.2 | 178 |

## 4.3 COMPUTATIONAL ANALYSIS

we have added a detailed comparison of training time, fusion time, and memory usage across baselines, including HOT-CLIP, Direct OT Fusion, and FARE. All GPU hours are measured on NVIDIA RTX 4090 GPUs. The results are summarized in Table 4.

In the training phase, our method involves fine-tuning 9 sub-models for 2 epochs on 1M images with 10-step adversarial attacks (e.g., PGD). This represents only about 1.4% of the computational cost of training the original CLIP model (trained for 32 epochs on 400M images). Moreover, the training of all sub-models is fully parallelizable, so with sufficient resources, the wall-clock time can be reduced to that of training a single sub-model. We believe this one-time training investment is both manageable and justified, as it produces a single and robust model that supports efficient inference.

In the fusion phase, the hierarchical structure itself acts as an optimization, effectively reducing memory requirements from $O(|\mathcal{A}||\mathcal{T}| \cdot U)$ to $O(\max\{|\mathcal{A}|, |\mathcal{T}|\} \cdot U)$ (the details are analyzed in Appendix C.3). We also employ the Sinkhorn algorithm to efficiently approximate the optimal transport solutions, significantly accelerating the computation while maintaining stability. In our experiments, each fusion stage—either intra-attack or inter-attack—typically completes in approximately 20 minutes, with peak memory usage reduced from 641GB to 178GB. We consider this computational overhead to be reasonable in practice, given the resulting robustness and inference efficiency.

## 5 CONCLUSION AND FUTURE WORK

The proposed HOT-CLIP framework enhances the adversarial robustness of large vision-language models by constructing diverse submodels across different adversarial attacks and textual prompts, and integrating them via hierarchical OT Fusion. Experiments on zero-shot image classification, VQA, and image captioning demonstrate consistent improvements in robustness under strong adversarial perturbations, while preserving competitive performance on clean data. Although leveraging multiple submodels increases computational cost during training, the hierarchical fusion ensures that inference-time overhead remains comparable to a single backbone, making the method practical for deployment. We think a promising direction for future work is to extend HOT-CLIP towards cross-modal joint defenses, examining the potential of hierarchical fusion for the simultaneous enhancement of robustness across multiple modalities.

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

## A    THE USE OF LLMs

A large language model (LLM) was employed for language polishing and grammar correction. All scientific ideas, experimental design, analysis, and conclusions were generated solely by the authors. The LLM did not contribute to any research ideation or content creation.

## B    RELATED WORKS

In this section, we review three main lines of research related to our study: large vision-language models (LVLMs), adversarial robustness of LVLMs, and optimal transport-based fusion methods.

**Large Vision-Language Models** In recent years, a number of large vision-language models have been released, demonstrating the rapid progress of multimodal learning. Representative open-source efforts include Flamingo (Alayrac et al., 2022), LLaVA (Liu et al., 2023), and more recently Qwen-VL(Bai et al., 2023) and InternVL (Chen et al., 2024), which extend large language models with visual perception capabilities. Despite architectural differences, most LVLMs adopt CLIP (Radford et al., 2021) or its variants as the vision-language alignment backbone. While these models provide transferable multimodal features for diverse downstream applications, they remain vulnerable to adversarial perturbations, raising concerns for deployment in safety-critical scenarios.

**Adversarial Robustness of LVLMs** Adversarial robustness in LVLMs has attracted increasing attention, as perturbations applied to visual, textual or jointly across both modalities can severely disrupt cross-modal representations. In this work, we focus on adversarial perturbations in the visual modality. Since the high-dimensional and continuous nature of visual data (Ye et al., 2025), and the strong transferability of visual adversarial examples (Waseda et al., 2023), make defending against visual attacks particularly challenging. Furthermore, in many LVLMs applications (such as medical imaging (Javed et al., 2024) and autonomous driving (Rossolini et al., 2023)), visual inputs often serve as the primary source of information for model decision-making. Existing defense strategies against visual modalities attacks can be grouped into two main categories: inference-phase defenses and training-phase defenses (Ye et al., 2025). Inference-phase defenses mitigate vulnerabilities at deployment time, typically by perturbing images before model inference (Xu et al., 2024; Sun et al., 2024). These approaches are attractive for their plug-and-play nature, but often cannot fundamentally eliminate the vulnerability of the vision encoder to adversarial perturbations. In contrast, training-phase defenses aim to improve robustness during model development, most commonly through adversarial fine-tuning (Schlarmann et al., 2024; Hossain & Imteaj, 2024b;a). While effective against certain attacks, such methods often struggle to generalize to unseen perturbations due to overfitting to specific attack (Rice et al., 2020). Our work falls within the scope of training-phase defenses, extending adversarial fine-tuning with an optimal transport–based fusion mechanism. Our proposed approach belongs to the category of training-phase defenses, extending adversarial fine-tuning with an optimal transport–based fusion mechanism.

**Optimal Transport Fusion** OT Fusion is a layer-wise model fusion technique that utilizes optimal transport to align neurons across the models before averaging their associated parameters (Singh & Jaggi, 2020). This technique has recently been extended to a variety of architectures, including transformers (Imfeld et al., 2024) and graph neural networks (Ormaniec et al., 2025). Beyond architectural adaptations, FedSKF (Zhou & Wang, 2024) introduces OT Fusion for knowledge integration in federated class-incremental learning, which aligns feature distributions between client and server models to mitigate data heterogeneity. Despite its success in model integration, OT Fusion has seen limited investigation in the context of adversarial robustness, which motivates our study.

**Optimal Transport (OT)** OT has been widely adopted in various alignment tasks, including document alignment Wang et al. (2024) and word alignment Arase et al. (2023). Melnyk et al. (2024) proposed AOT, an OT-based framework that aligns reward distributions for large language models by enforcing distributional preference dominance. OT also offers new perspectives on existing techniques. For example, recent studies (Shi et al., 2023; 2024) show that Contrastive Learning (CL) and the CLIP model can be reformulated as (Inverse) OT problems, where common objectives such as InfoNCE loss can be interpreted as instances of (Inverse) OT for aligning sample similarities.

---

**Algorithm 1** Hierarchical OT Fusion for CLIP Visual Encoder

---

**Require:** Clean dataset $\mathcal{D} = \{(x, y)\}$; attack set $\mathcal{A}$; template set $\mathcal{T}$; standard template $t_0$; layer indices $\ell = 1 \dots L$.

**Ensure:** Fused visual encoder $\bar{f}_{\theta_{\text{img}}}$ (text encoder frozen).

1: **for** each $a \in \mathcal{A}$ **do**
2:     **for** each $t \in \mathcal{T}$ **do**
3:         Construct diversified dataset: $\mathcal{D}_{\text{div}}^{(a,t)} \leftarrow \{(x + \delta_a(x), y^{(t)}) : (x, y) \in \mathcal{D}\}$.
4:         Train submodel $M^{(a,t)} \leftarrow \text{AdversarialTrain}(\mathcal{D}_{\text{div}}^{(a,t)})$ with frozen text encoder.
5:         Store layer weights $\{W_\ell^{(a,t)}\}_{\ell=1}^L$.
6:     **end for**
7: **end for**
8: **Level 1: Prompt-level fusion within each attack**
9: **for** each $a \in \mathcal{A}$ **do**
10:     Select anchor model $M^{(a, t_{\text{anchor}})}$.
11:     **for** $\ell = 1$ to $L$ **do**
12:         **for** each $t \in \mathcal{T}$ **do**
13:             Align columns of weights: $\tilde{W}_\ell^{(a,t)} \leftarrow W_\ell^{(a,t)} T_{\ell-1}^{(a,t)}$
14:             Compute OT plan: $T_\ell^{(a,t)} \leftarrow \text{ComputeOT}(\tilde{W}_\ell^{(a,t)}, W_\ell^{(a, t_{\text{anchor}})})$.
15:             Align weights: $W_\ell^{(a,t),\text{aligned}} \leftarrow (T_\ell^{(a,t)})^\top W_\ell^{(a,t)} T_{\ell-1}^{(a,t)}$.
16:         **end for**
17:         Average aligned weights: $W_\ell^{(a)} \leftarrow \frac{1}{|\mathcal{T}|} \sum_{t \in \mathcal{T}} W_\ell^{(a,t),\text{aligned}}$.
18:     **end for**
19:     Assemble fused model $\bar{M}^{(a)}$ with $\{W_\ell^{(a)}\}$.
20:     Fine-tune $\bar{M}^{(a)}$ on $\mathcal{D}_{\text{div}}^{(a,t_0)}$.
21: **end for**
22: **Level 2: Attack-level fusion across attacks**
23: Select anchor attack $a_{\text{anchor}}$.
24: **for** $\ell = 1$ to $L$ **do**
25:     **for** each $a \in \mathcal{A}$ **do**
26:         Align columns of weights: $\tilde{W}_\ell^{(a)} \leftarrow W_\ell^{(a)} T_{\ell-1}^{(a)}$
27:         Compute OT plan: $T_\ell^{(a)} \leftarrow \text{ComputeOT}(\tilde{W}_\ell^{(a)}, W_\ell^{(a_{\text{anchor}})})$.
28:         $W_\ell^{(a),\text{aligned}} \leftarrow (T_\ell^{(a)})^\top W_\ell^{(a)} T_{\ell-1}^{(a)}$.
29:     **end for**
30:     $W_\ell^{\text{fused}} \leftarrow \frac{1}{|\mathcal{A}|} \sum_{a \in \mathcal{A}} W_\ell^{(a),\text{aligned}}$.
31: **end for**
32: Assemble final fused model $\bar{f}_{\theta_{\text{img}}}$ with $\{W_\ell^{\text{fused}}\}_{\ell=1}^L$.
33: **return** $\bar{f}_{\theta_{\text{img}}}$.

---

# C   Algorithm

In this section, we provide the detailed procedure of our HOT-CLIP in Algorithm 1. Concretely, we first adversarially train a diverse set of submodels, each under a specific attack method and textual template. Within each attack family, the submodels are aligned at the neuron level using optimal transport maps and then averaged to obtain an intra-attack fused model, which is further fine-tuned under the corresponding attack setting. (The procedure relies on computing optimal transport maps through the subroutine ComputeOT(.), whose implementation details are provided in Appendix C.1.) In the second stage, the intra-attack fused models are again aligned and averaged across different attacks, followed by an additional round of fine-tuning to adapt the final visual encoder. This step-by-step process complements the high-level overview in the main text by making the construction, alignment, and fusion operations explicit.

## C.1 DETAILS OF OPTIMAL TRANSPORT FUSION

In this section, we introduce the details of OT Fusion (Singh & Jaggi, 2020) and Transformer-specific OT Fusion (Imfeld et al., 2024), following the methodologies of Singh & Jaggi (2020) and Imfeld et al. (2024).

### C.1.1 OPTIMAL TRANSPORT FUSION

We provide a more formal description of the OT Fusion procedure. Consider two submodels $A$ and $B$, and suppose we are at layer $\ell$, with neurons in previous layers already aligned.

**Step 1: Define probability measures.** We define probability measures over neurons at layer $\ell$ for the two models as $\mu^{(\ell)} = \left(\alpha^{(\ell)}, X^{(\ell)}\right)$, $\nu^{(\ell)} = \left(\beta^{(\ell)}, Y^{(\ell)}\right)$, where $X^{(\ell)} = \{x_1^{(\ell)}, \ldots, x_{n^{(\ell)}}^{(\ell)}\}$ and $Y^{(\ell)} = \{y_1^{(\ell)}, \ldots, y_{m^{(\ell)}}^{(\ell)}\}$ are the neuron weight vectors of models $A$ and $B$. We use uniform histograms as initialization: $\alpha^{(\ell)} \leftarrow \frac{1}{n^{(\ell)}} \mathbf{1}_{n^{(\ell)}}$, $\beta^{(\ell)} \leftarrow \frac{1}{m^{(\ell)}} \mathbf{1}_{m^{(\ell)}}$.

**Step 2: Propagate alignment from previous layer.** To ensure consistency, we first align the incoming edge weights for layer $\ell$ using the transport plan from the previous layer, $T^{(\ell-1)}$. Formally,

$$\widetilde{W}_A^{(\ell,\ell-1)} \leftarrow W_A^{(\ell,\ell-1)} T^{(\ell-1)} \operatorname{diag}\left(\frac{1}{\beta^{(\ell-1)}}\right), \tag{14}$$

where $W_A^{(\ell,\ell-1)}$ is the weight matrix between layers $\ell - 1$ and $\ell$ in model $A$. This step aligns the current layer's weights, $W_A^{(\ell,\ell-1)}$, based on the transport map of the preceding layer, so that the subsequent computation of the optimal transport map for this layer is meaningful.

**Step 3: Solve optimal transport at current layer.** Given a ground cost $D_S(\cdot, \cdot)$ (we use Euclidean distance), the transport plan $T^{(\ell)}$ is obtained by solving

$$T^{(\ell)} \leftarrow \operatorname{OT}\left(\mu^{(\ell)}, \nu^{(\ell)}; D_S\right), \tag{15}$$

We solve the OT problem for the current layer using the Sinkhorn algorithm (Cuturi, 2013)

**Step 4: Align neurons and fuse.** Using $T^{(\ell)}$, we align model $A$'s weights with respect to model $B$:

$$W_{A,\text{aligned}}^{(\ell,\ell-1)} \leftarrow \operatorname{diag}(1/\beta^{(\ell)}) T^{(\ell)\top} \widetilde{W}_A^{(\ell,\ell-1)}. \tag{16}$$

Finally, the fused weights are obtained as

$$W_F^{(\ell,\ell-1)} \leftarrow \tfrac{1}{2}\left(W_{A,\text{aligned}}^{(\ell,\ell-1)} + W_B^{(\ell,\ell-1)}\right). \tag{17}$$

Then, the above procedure is applied sequentially across all layers.

### C.1.2 TRANSFORMER FUSION

In Transformers, the flow of transportation maps (TMs) becomes more complex due to structure of Transformers (residual connections, multi-head attention, and normalization layers). In this section, we introduce the OT Fusion of transformer (Imfeld et al., 2024).

**Residual Connections.** For a residual block, the outputs of the main branch and the skip branch are summed. Thus, the outgoing TM depends on both incoming TMs. We use a weighted averaging strategy:

$$T_{\text{out}}^{(\ell)} = T_{\text{main}}^{(\ell)} \operatorname{diag}(1 - \gamma^{(\ell)}) + T_{\text{res}}^{(\ell)} \operatorname{diag}(\gamma^{(\ell)}), \tag{18}$$

where $\gamma^{(\ell)}$ is a weighting vector that controls the relative contribution of the main branch and the skip (residual) branch.

**Multi-Head Attention.** Multi-head attention introduces additional challenges for OT Fusion, as transport maps must be propagated consistently across the query ($W_Q$), key ($W_K$), value ($W_V$), and output ($W_O$) projections. Recall that the attention mechanism is defined as

$$\text{Attention}(Q, K, V) = \text{softmax}\left(\frac{QK^\top}{\sqrt{d_k}}\right) V, \tag{19}$$

where $Q = XW_Q$, $K = XW_K$, and $V = XW_V$. We adopt the following alignment strategy: **(i) Propagation across $Q$, $K$, and $V$:** The transport maps for $W_Q$, $W_K$, and $W_V$ are inherited directly from the previous layer, which applies equally to the multi-head case. **(ii) Handling $W_O$ under hard alignment:** The output projection $W_O$ depends jointly on the aligned $Q$, $K$, and $V$ branches. Under hard alignment, we enforce $T_Q = T_K = T_{QK}$ so that the permutation cancels inside the softmax operation, leaving the attention scores unchanged:

$$\text{softmax}\left(\frac{(QT_Q)(KT_Q)^\top}{\sqrt{d_k}}\right) = \text{softmax}\left(\frac{QK^\top}{\sqrt{d_k}}\right). \tag{20}$$

In this case, only the transport map of $V$ is propagated to $W_O$. **(iii) Cross-head alignment:** Because there is no guarantee of one-to-one correspondence between heads across models, we adopt a cross-head alignment strategy. Specifically, the projection matrices for each head $\{W_Q^i, W_K^i, W_V^i\}$ are concatenated across all heads to form unified matrices $W_Q, W_K, W_V$. OT Fusion is then applied to these concatenated matrices. Finally, the transport map $T_V$ is propagated to $W_O$.

**Feed-Forward Networks, Layer Normalization, and Embeddings.** Each feed-forward sublayer is treated as a standard linear layer. Layer normalization contains only per-dimension affine parameters ($\alpha, \beta$), which are aligned directly using the incoming transport map. For positional embeddings, which are added residually, we apply the same fusion strategies as used for residual connections.

## C.2 Proof of lemma 3.1

In this section, we provide the proof of lemma 3.1

*Proof.* To understand why the hierarchical barycenter $\mu_{\text{hier}}^*$ is close to the global barycenter $\mu_{\text{global}}^*$, let's reason step by step.

First, recall that the triangle inequality allows us to break a distance into two parts. Intuitively, the distance between the hierarchical barycenter and the global barycenter can be bounded by the distance from the hierarchical barycenter to each group barycenter, plus the distance from each group barycenter to the global barycenter:

$$W_c(\mu_{\text{hier}}^*, \mu_{\text{global}}^*) \leq \frac{1}{|\mathcal{A}|} \sum_a \left[ W_c(\mu_{\text{hier}}^*, \mu_a^*) + W_c(\mu_a^*, \mu_{\text{global}}^*) \right]. \tag{21}$$

Next, we use the key property of a barycenter: by definition, $\mu_{\text{hier}}^*$ minimizes the average distance to all group barycenters. This implies that the sum of distances from $\mu_{\text{hier}}^*$ to each $\mu_a^*$ is no larger than the sum of distances from the global barycenter to each $\mu_a^*$:

$$\sum_a W_c(\mu_{\text{hier}}^*, \mu_a^*) \leq \sum_a W_c(\mu_{\text{global}}^*, \mu_a^*). \tag{22}$$

Plugging this into the previous inequality gives a first bound on the hierarchical-global distance:

$$W_c(\mu_{\text{hier}}^*, \mu_{\text{global}}^*) \leq \frac{2}{|\mathcal{A}|} \sum_a W_c(\mu_a^*, \mu_{\text{global}}^*). \tag{23}$$

Now we examine each group barycenter $\mu_a^*$. Similarly, applying the triangle inequality at this intra-group level gives:

$$W_c(\mu_a^*, \mu_{\text{global}}^*) \leq \frac{1}{|\mathcal{T}|} \sum_t \left[ W_c(\mu_a^*, \mu_{a,t}) + W_c(\mu_{a,t}, \mu_{\text{global}}^*) \right]. \tag{24}$$

Combining (24) with (23), we propagate the bound from the group level to the individual submodels:

$$W_c(\mu_{\text{hier}}^*, \mu_{\text{global}}^*) \leq \frac{2}{|\mathcal{AT}|} \sum_a \sum_t \Big[ W_c(\mu_a^*, \mu_{a,t}) + W_c(\mu_{a,t}, \mu_{\text{global}}^*) \Big]. \tag{25}$$

Finally, by the barycenter property within each group, the distance from the group barycenter to its submodels is at most the distance from the submodels to the global barycenter:

$$\sum_t W_c(\mu_a^*, \mu_{a,t}) \leq \sum_t W_c(\mu_{\text{global}}^*, \mu_{a,t}). \tag{26}$$

Substituting this inequality, we obtain a simple bound that directly relates the hierarchical barycenter to all individual submodels:

$$W_c(\mu_{\text{hier}}^*, \mu_{\text{global}}^*) \leq \frac{4}{|\mathcal{AT}|} \sum_{a,t} W_c(\mu_{a,t}, \mu_{\text{global}}^*), \tag{27}$$

which intuitively shows that if each submodel is close to the global barycenter, then the hierarchical OT fusion will also remain close.

$\square$

## C.3 ANALYSIS OF MEMORY USAGE

Let $|\mathcal{A}|$ denote the number of adversarial attack types, $|\mathcal{T}|$ denote the number of textual prompts and $U$ represent the memory usage of a single submodel. According to the method of align to an anchor model, In naive global fusion, all $|\mathcal{A}| \cdot |\mathcal{T}|$ submodels are aligned and stored simultaneously, resulting in a peak memory complexity of $\mathcal{O}(|\mathcal{A}||\mathcal{T}| \cdot U)$.

In our two-level hierarchical OT Fusion framework, fusion is performed progressively: **Intra-attack fusion (L-level):** For each attack $a \in \mathcal{A}$, only the $|\mathcal{T}|$ submodels corresponding to different prompts are fused at a time. Once fused, the intermediate result replaces the original submodels, so the memory required at any step is proportional to $\{|\mathcal{T}|\}$. **Inter-attack fusion (V-level):** The first-level fused models, one per attack type, are further integrated. Again, the number of models stored simultaneously is bounded by $\{|\mathcal{A}|\}$.

Since the fusion is hierarchical and progressive, the peak memory usage at any step never exceeds $\mathcal{O}(\max\{|\mathcal{A}|, |\mathcal{T}|\} \cdot U)$.

# D DETAILS OF IMPLEMENTATION

In this section, we detail the implementation and training setup. All models use CLIP (Radford et al., 2021) with a ViT-L/14 visual encoder, initialized from the official OpenAI checkpoint on Hugging Face. Finetuning is performed on 8 NVIDIA RTX 4090 GPUs with PyTorch 2.5 and CUDA 12.4, using AdamW (Loshchilov & Hutter, 2019) with a learning rate of $1 \times 10^{-5}$, weight decay 0.01, and batch size 128 (16 per GPU). Each submodel is adversarially fine-tuned for 2 epochs on ImageNet under $\ell_\infty$ perturbations ($\epsilon = 4/255$). For each attack method (FGSM (Goodfellow et al., 2015), PGD (Madry et al., 2018), and MIM (Dong et al., 2018)), we train three submodels per attack, each using a textual prompt randomly sampled from 80 templates following standard CLIP practice (Radford et al., 2021), to ensure diverse adversarial examples. For OT Fusion, neurons in each linear layer are represented by their incoming weight vectors, which are $\ell_2$-normalized prior to computing pairwise distances. The ground cost is defined as the squared Euclidean distance. Transport matrices are computed using the Sinkhorn (Cuturi, 2013) algorithm in the stabilized log-domain, with entropic regularization $\tau = 8 \times 10^{-2}$. Each submodel is aligned to an anchor model trained with the standard template ("This is a photo of a "). During intra-attack fusion, the fused model is fine-tuned for 1 epoch with the corresponding attack and the standard template. During inter-attack fusion, the model undergoes 1 epoch of unsupervised adversarial fine-tuning, while reusing the same optimizer and learning rate schedule. A summary of all settings is provided in Table 5.

Table 5: Hyperparameters used in our experiments.

| Stage | Parameter | Value |
|---|---|---|
| Backbone & Setup | Visual Encoder | CLIP ViT-L/14 (OpenAI) |
| | Hardware | $8\times$ NVIDIA RTX 4090 GPUs |
| | Framework | PyTorch 2.5,CUDA 12.4 |
| | Batch Size | 128 (16 per GPU) |
| Adversarial Training | Epochs | 2 |
| | Dataset | ImageNet |
| | Attacks | FGSM, PGD, MIM |
| | Perturbation | $\ell_\infty, \epsilon = 4/255$ |
| | Optimizer | AdamW |
| | Learning Rate | $1 \times 10^{-5}$ |
| | Weight Decay | 0.01 |
| OT Fusion | Representation | weight vectors |
| | Solver | Sinkhorn ( $\tau = 8 \times 10^{-2}$) |
| | Ground metric | squared Euclidean distance |
| | Measure | uniform |

We evaluate our models across three tasks: image classification, image captioning, and visual question answering (VQA), considering both clean performance and adversarial robustness. Across all tasks, we adopt AutoAttack (Croce & Hein, 2020) under the $\ell_\infty$ norm with perturbation radii $\epsilon = 2/255$ and $\epsilon = 4/255$, each run for 100 iterations. This provides a standardized and reliable evaluation of robustness against adversarial perturbations.

**Image classification.** This task requires assigning a semantic label to each input image. We evaluate clean and robust accuracy on ImageNet and 13 additional zero-shot datasets (details in Section 4.1). For each dataset, class names are combined with a predefined set of prompt templates, and zero-shot classification is performed as described in Section 2. Accuracy is reported as the proportion of correctly classified images.

**Image captioning.** Here, the model generates natural language descriptions conditioned on an image. We evaluate performance on the COCO (Lin et al., 2014) and Flickr30k (Young et al., 2014) datasets using the CIDEr score (Vedantam et al., 2015), a consensus-based metric that measures the similarity of generated captions to human-written references. For image captioning, OpenFlamingo-9B (OF) (Alayrac et al., 2022) is evaluated in a zero-shot setting without additional in-context exemplars, whereas LLaVA-1.5-7B (Liu et al., 2023) is evaluated using its default system prompt and captioning prompt. Clean evaluation uses the full validation sets, and adversarial evaluation is conducted on 500 randomly sampled images per dataset, following Schlarmann & Hein (2023).

**Visual question answering.** This task requires the model to answer natural language questions based on an image. We evaluate on two widely used benchmarks, VQAv2 (Goyal et al., 2017) and TextVQA (Singh et al., 2019). Performance is measured by VQA accuracy, which computes the proportion of model predictions that match human-annotated answers, thereby reflecting both linguistic and visual reasoning ability. The same LVLMs and evaluation settings as in the captioning task are used.

**Datasets.** (1) For zero-shot image classification, we use ImageNet (Deng et al., 2009), Caltech-101 (Fei-Fei et al., 2007), Cars (Krause et al., 2013), CIFAR-10 (Krizhevsky et al., 2009), CIFAR-100 (Krizhevsky et al., 2009), DTD (Cimpoi et al., 2014), EuroSAT (Helber et al., 2019), FGVC Aircraft (Maji et al., 2013), Flowers (Nilsback & Zisserman, 2008), ImageNet-R (Hendrycks et al., 2021), ImageNet-S (Gao et al., 2023), PCAM (Veeling et al., 2018), Oxford Pets (Parkhi et al., 2012), STL-10 (Coates et al., 2011). (2) For visual question answering (VQA), we use TextVQA (Singh et al., 2019) and VQAv2 (Goyal et al., 2017). (3) For image captioning, we use COCO (Lin et al., 2014) and Flickr30k (Young et al., 2014). For classification and VQA, we report the accuracies under clean and adversarial inputs. For image captioning, we use CIDEr (Vedantam et al., 2015) to evaluate the quality of generated captions under attack.

Table 6: Comparison of fusion strategies on zero-shot image classification. Robustness is measured using AutoAttack with $\ell_\infty$ perturbations bounded by $\epsilon = 2/255$ and $\epsilon = 4/255$. Bold numbers indicate the best performance in each column.

| | strategy | ImageNet | CalTech | Cars | CIFAR10 | CIFAR100 | DTD | EuroSAT | FGVC | Flowers | ImageNet-R | ImageNet-S | PCAM | OxfordPets | STL-10 |
|---|---|---|---|---|---|---|---|---|---|---|---|---|---|---|---|
| clean | Direct Average | 69.3 | **85.6** | 63.9 | 76.4 | 58.7 | 44.5 | 17.9 | **23.0** | **57.0** | 81.3 | **60.0** | 49.5 | 86.2 | 96.1 |
| | Direct OTFusion | 69.2 | 85.0 | **65.0** | 76.4 | 58.6 | 44.7 | 17.6 | 22.5 | 56.8 | **82.0** | 59.7 | 49.5 | 86.2 | 96.2 |
| | HOT-CLIP | **69.9** | 85.1 | 64.0 | **78.7** | **60.0** | **46.2** | **18.0** | 21.8 | 56.9 | 80.8 | 59.9 | **49.6** | **87.1** | **96.3** |
| $\epsilon = 2/255$ | Direct Average | 52.8 | 78.2 | 30.0 | 56.9 | 36.1 | 30.6 | **12.5** | 8.0 | 27.8 | 60.5 | 43.2 | 49.5 | 70.9 | 90.0 |
| | Direct OTFusion | 52.5 | 78.4 | 30.2 | 56.2 | 35.7 | 30.7 | 12.1 | 8.0 | 28.1 | 60.2 | 43.6 | 49.5 | 71.1 | 90.1 |
| | **HOT-CLIP** | **53.4** | **79.5** | **31.8** | **58.3** | **37.2** | **31.8** | 12.4 | **8.1** | **29.7** | **60.9** | **44.6** | **49.6** | **71.2** | **91.0** |
| $\epsilon = 4/255$ | Direct Average | 31.8 | 63.7 | 15.1 | 36.2 | 19.9 | 19.2 | 9.6 | 2.0 | 12.2 | 39.0 | 31.6 | 49.5 | 51.0 | 74.9 |
| | Direct OTFusion | 32.0 | 64.3 | 15.1 | 36.3 | 20.4 | 19.0 | 9.5 | 2.0 | 12.0 | 38.9 | 30.5 | 49.5 | 51.0 | 75.0 |
| | **HOT-CLIP** | **34.7** | **66.5** | **15.8** | **38.1** | **20.9** | **19.6** | **10.3** | **2.9** | **12.5** | **39.4** | **32.8** | **49.6** | **51.8** | **77.0** |

Table 7: Comparison of fusion strategies on image caption and VQA. Robustness is measured using AutoAttack with $\ell_\infty$ perturbations bounded by $\epsilon = 2/255$ and $\epsilon = 4/255$. Bold numbers indicate the best performance in each column..

| | VLM strategy | COCO | | | Flickr30 | | | TextVQA | | | VQAv2 | | |
|---|---|---|---|---|---|---|---|---|---|---|---|---|---|
| | | clean | $\epsilon=\frac{2}{255}$ | $\epsilon=\frac{4}{255}$ | clean | $\epsilon=\frac{2}{255}$ | $\epsilon=\frac{4}{255}$ | clean | $\epsilon=\frac{2}{255}$ | $\epsilon=\frac{4}{255}$ | clean | $\epsilon=\frac{2}{255}$ | $\epsilon=\frac{4}{255}$ |
| LLaVA | Direct Average | **114.7** | 46.6 | 30.8 | 73.9 | 36.7 | 26.1 | 24.7 | 18.4 | 9.8 | 67.8 | **44.2** | 31.4 |
| | Direct OTFusion | 113.5 | 47.7 | 30.9 | 73.7 | 37.2 | 25.0 | 25.3 | **18.7** | 8.8 | 67.8 | 43.2 | 31.3 |
| | HOT-CLIP | 110.4 | **56.5** | **35.5** | **74.6** | **43.1** | **26.5** | 25.3 | 17.7 | **12.8** | **68.8** | 43.8 | **34.6** |

# E    ADDITIONAL EXPERIMENTS RESULTS

## E.1    ABLATION STUDIES

**Ablation on Fusion Strategies.** To evaluate the effectiveness of our hierarchical OT Fusion design, we compare three different fusion strategies on three vision-language tasks: zero-shot image classification, image captioning, and visual question answering. The first strategy, Direct Average, simply averages the parameters of all submodels without any alignment. The second strategy, Direct OT Fusion, applies optimal transport to align all submodels simultaneously before averaging. The third strategy, Hierarchical OT Fusion (HOT-CLIP), performs two-level fusion: intra-attack alignment followed by inter-attack integration. For each task, we report both clean and robust performance, with robustness evaluated using AutoAttack under the $\ell_\infty$ norm with $\epsilon = 2/255$. Results in Table 6 and Table 7 indicate that Direct Average and Direct OT Fusion often suffers from misalignment across diverse submodels, leading to degraded clean accuracy and limited robustness gains. In contrast, HOT-CLIP consistently improves robustness while maintaining or slightly enhancing clean performance. This highlights the advantage of controlling submodel similarity at each fusion stage, demonstrating that hierarchical fusion is crucial for effectively leveraging diverse adversarially trained submodels.

**Ablation on the Number of Submodels** Following the fusion tratategy, we investigate how the size and composition of the submodel pool affect the performance, we vary: the number of families $K$ (corresponding to distinct attack methods), and the number of submodels per family $J$ (corresponding to different prompt variants per attack). We examine several configurations: $3 \times 3$, $2 \times 3$, $1 \times 3$, and $3 \times 1$, where the first number denotes $K$ and the second $J$. All other settings follow Section 5.1 unless specified otherwise. We report CIDEr score and VQA accuracy for image captioning

Table 8: Clean and adversarial zero-shot performance on Image Captioning and VQA for different submodel pool configurations.

| VLM | Num. | COCO | | | Flickr30 | | | TextVQA | | | VQAv2 | | |
|---|---|---|---|---|---|---|---|---|---|---|---|---|---|
| | | clean | $\epsilon = \frac{2}{255}$ | $\epsilon = \frac{4}{255}$ | clean | $\epsilon = \frac{2}{255}$ | $\epsilon = \frac{4}{255}$ | clean | $\epsilon = \frac{2}{255}$ | $\epsilon = \frac{4}{255}$ | clean | $\epsilon = \frac{2}{255}$ | $\epsilon = \frac{4}{255}$ |
| LLaVA | $(1 \times 3)$ | 108.4 | 49.6 | 30.1 | 74.4 | 40.0 | 26.6 | 25.3 | 17.7 | 8.8 | 67.8 | **44.9** | 31.3 |
| | $(2 \times 3)$ | **113.6** | 49.7 | 30.9 | 73.7 | 37.2 | 25.0 | 25.0 | **18.7** | 8.8 | 67.9 | 43.2 | 31.4 |
| | $(3 \times 3)$ | 110.4 | **56.5** | **35.5** | **74.6** | **43.1** | 26.5 | 25.3 | 17.7 | **12.8** | **68.8** | 43.8 | **34.6** |
| | $(3 \times 1)$ | 110.6 | 52.2 | 32.3 | 74.5 | 39.2 | **27.3** | **26.3** | 17.7 | 9.2 | 67.8 | 44.2 | 31.9 |

Table 9: Comparison of HOT-CLIP with different adversarial training strengths on image captioning and VQA tasks.

| VLM | Strength | COCO | | | Flickr30 | | | TextVQA | | | VQAv2 | | |
|---|---|---|---|---|---|---|---|---|---|---|---|---|---|
| | | clean | $\epsilon = \frac{2}{255}$ | $\epsilon = \frac{4}{255}$ | clean | $\epsilon = \frac{2}{255}$ | $\epsilon = \frac{4}{255}$ | clean | $\epsilon = \frac{2}{255}$ | $\epsilon = \frac{4}{255}$ | clean | $\epsilon = \frac{2}{255}$ | $\epsilon = \frac{4}{255}$ |
| LLaVA | 2/255 | **111.9** | 48.1 | 33.0 | 73.7 | **43.1** | 24.0 | **25.3** | **17.7** | 9.8 | 66.8 | **44.9** | 32.2 |
| | 4/255 | 110.4 | **56.5** | **35.5** | **74.6** | **43.1** | **26.5** | **25.3** | **17.7** | **12.8** | **68.8** | 43.8 | **34.6** |

and VQA tasks, respectively. As shown in Table 8, increasing the number of families $K$ generally improves robustness, indicating the benefit of incorporating diverse attack types. Adding more sub-models per family ($J$) also enhances performance, demonstrating that prompt diversity contributes to more robust feature representations. Notably, the $3 \times 3$ configuration achieves the highest overall gains across all tasks.

**Ablation on Adversarial Training Strength.** We further study the impact of adversarial training strength on the robustness of our approach. Specifically, we train submodels with perturbation radii $\epsilon_{\text{train}} \in \{2/255, 4/255\}$ under the $\ell_\infty$ norm, keeping all other training settings fixed. Evaluation is conducted on ImageNet using AutoAttack at $\epsilon_{\text{eval}} = 2/255$ and $\epsilon_{\text{eval}} = 4/255$, in addition to clean accuracy. As shown in Table 9, models adversarially trained with $\epsilon_{\text{train}} = 2/255$ achieve higher clean performance but are less robust under stronger attacks, while those trained with $\epsilon_{\text{train}} = 4/255$ exhibit the opposite trend.

**Ablation Studies on Visual Encoder Backbone.** To evaluate the general applicability of our HOT-CLIP framework, we replace the CLIP ViT-L/14 visual encoder with the smaller CLIP ViT-B/32, while keeping all other training and hierarchical OT Fusion settings unchanged. The resulting models are evaluated on zero-shot image classification. As shown in Table 10, models using ViT-B/32 achieve lower absolute accuracy than ViT-L/14 due to reduced capacity (e.g., 63.5% vs 71.2% on ImageNet). Nonetheless, our hierarchical OT Fusion consistently improves robustness, increasing accuracy by approximately 4.2% over the ViT-B/32 baseline and 3.8% over the ViT-L/14 baseline under adversarial evaluation. These results indicate that HOT-CLIP effectively generalizes across different visual encoder backbones while providing tangible robustness gains.

**Extended Robustness Evaluation Across Models, Attacks, and Norms** To more comprehensively evaluate the generality and stability of our proposed method, we extend our analysis beyond the primary setup in the main paper. Specifically, we conduct three additional sets of experiments to examine whether our approach remains effective across different model architectures, attack algorithms, and perturbation metrics. We first apply our method to the BLIP2 (Li et al., 2023a) architecture by replacing its default vision encoder with our HOT-CLIP. We evaluate performance on COCO and Flickr30k image captioning, as well as TextVQA and VQAv2. As shown in Table 11, HOT-CLIP provides consistent robustness improvements, demonstrating that our framework is compatible with LVLMs beyond CLIP-based architectures. To further assess the robustness of our method under different adversarial attack, we additionally evaluate the models using PGD attacks. The results in Table 12 show that our method consistently improves robustness under both AutoAttack and PGD, confirming the stability of our approach across different attack algorithms. Finally, we evaluate robustness under $L_2$-bounded AutoAttacks using two perturbation budgets ($\epsilon = 0.5$ and $\epsilon = 1.0$). As

Table 10: Zero-shot classification performance and adversarial robustness of CLIP models visual encoder ViT-B/32. Robustness is measured using AutoAttack with $\ell_\infty$ perturbations bounded by $\epsilon = 2/255$.

| Eval. | Vision encoder | ImageNet | CalTech | Cars | CIFAR10 | CIFAR100 | DTD | EuroSAT | FGVC | Flowers | ImageNet-R | ImageNet-S | PCAM | OxfordPets | STL-10 | Avg. |
|---|---|---|---|---|---|---|---|---|---|---|---|---|---|---|---|---|
| clean | CLIP | 59.3 | 82.1 | 60.8 | 89.2 | 59.1 | 46.3 | 53.6 | 20.3 | 68.5 | 65.6 | 41.2 | 62.8 | 88.9 | 97.5 | 63.9 |
| | TeCoA | 55.9 | 71.4 | 14.3 | 70.6 | 40.2 | 26.7 | 20.1 | 6.0 | 27.3 | 48.3 | 27.5 | 49.4 | 72.0 | 86.9 | 44.0 |
| | FARE | 48.7 | 80.6 | 34.2 | 68.1 | 46.0 | 34.7 | 16.0 | 11.2 | 38.0 | 48.3 | 34.3 | 49.4 | 78.9 | 89.2 | 48.4 |
| | HOT-CLIP | 50.3 | 80.8 | 34.9 | 70.7 | 47.2 | 35.4 | 16.6 | 11.7 | 36.5 | 50.8 | 34.3 | 49.5 | 79.0 | 90.3 | 49.1 |
| $\epsilon = 2/255$ | CLIP | 0.0 | 0.0 | 0.0 | 0.0 | 0.0 | 0.0 | 0.0 | 0.0 | 0.0 | 0.0 | 0.0 | 0.0 | 0.0 | 0.0 | 0.0 |
| | TeCoA | **39.1** | 64.0 | 5.0 | 49.2 | 25.7 | 19.5 | 13.6 | 2.0 | 13.0 | **31.2** | 18.9 | 49.1 | **56.0** | 74.3 | 32.9 |
| | FARE | 29.6 | 67.0 | **12.9** | 47.2 | 27.6 | 22.8 | 12.9 | 3.7 | 14.1 | 29.2 | 20.8 | 49.4 | 50.0 | 77.3 | 33.1 |
| | HOT-CLIP | 30.6 | **67.1** | 11.5 | **49.5** | **29.0** | **24.3** | **13.7** | **3.8** | **15.2** | **31.2** | **22.5** | **49.6** | 51.8 | **77.4** | **34.0** |

Table 11: **Evaluation of BLIP2 with different encoders under AutoAttack.** Results are reported for image captioning (CIDEr) on COCO and Flickr30k, and VQA accuracy (%) on TextVQA and VQAv2. $\epsilon$ indicates the $l_\infty$ perturbation bound.

| VLM | Vision encoder | COCO | | | Flickr30 | | | TextVQA | | | VQAv2 | | |
|---|---|---|---|---|---|---|---|---|---|---|---|---|---|
| | | clean | $\epsilon=\frac{2}{255}$ | $\epsilon=\frac{4}{255}$ | clean | $\epsilon=\frac{2}{255}$ | $\epsilon=\frac{4}{255}$ | clean | $\epsilon=\frac{2}{255}$ | $\epsilon=\frac{4}{255}$ | clean | $\epsilon=\frac{2}{255}$ | $\epsilon=\frac{4}{255}$ |
| BLIP-2 | ViT-L/14 | **127.7** | 3.9 | 2.5 | **83.3** | 2.1 | 1.4 | **32.3** | 0.0 | 0.0 | **48.4** | 3.2 | 0.5 |
| | TeCoA | 99.7 | 45.2 | 23.1 | 53.6 | 30.7 | 19.8 | 24.5 | 8.1 | 5.2 | 45.8 | 25.6 | 20.1 |
| | FARE | 109.6 | 54.1 | 35.3 | 68.4 | 31.5 | 22.1 | 25.9 | 9.2 | 5.8 | 46.3 | 26.4 | 20.7 |
| | **HOT-CLIP** | 115.8 | **58.4** | **38.8** | 75.3 | **44.9** | **27.2** | 27.4 | **11.7** | **8.3** | 47.1 | **28.8** | **23.9** |

shown in Table 13, HOT-CLIP maintains strong robustness under $L_2$ perturbations, suggesting that the benefits of our approach generalize across multiple threat models.

## E.2 ADDITIONAL TASK

**Robustness under Stealthy Targeted Attacks.** Stealthy targeted attacks are high-risk adversarial scenarios, where the attacker aims to manipulate the model to produce a specific target output while the perturbation remains imperceptible to the user (Schlarmann & Hein, 2023). Such attacks pose real-world safety concerns, for example, by guiding users to phishing websites or spreading false information. To evaluate the effectiveness of HOT-CLIP under these conditions, we substitute the CLIP visual encoder in LLaVA-1.5 7B with our robust versions. We perform $\ell_\infty$ stealthy targeted attacks using APGD (Croce & Hein, 2020) with 500 iterations. Two perturbation radii are considered: $\epsilon = 2/255$ and $\epsilon = 4/255$. The target caption is "You are too stupid. I will not answer anymore" and is tested on 25 sampled images. An attack is deemed successful if the target string is exactly contained in the model's output. Table 14 reports the attack success rate (ASR) for different encoders. The original CLIP encoder is completely susceptible to attacks at $\epsilon = 2/255$, whereas HOT-CLIP achieves 0% ASR, demonstrating full robustness. Even at $\epsilon = 4/255$, HOT-CLIP substantially mitigates attacks compared to CLIP. Qualitative examples are shown in Fig. 4, illustrating that HOT-CLIP maintains high-quality captions while resisting the targeted manipulations. These findings extend our main results on general adversarial robustness, confirming that HOT-CLIP not only improves zero-shot classification, VQA, and image captioning robustness, but also protects LVLMs against realistic high-risk targeted attacks.

**Hallucination Experiments** Large vision-language models (LVLMs) are prone to object hallucination, where the model predicts the presence of objects that do not actually appear in the image.

Table 12: **Evaluation of LLaVA with different encoders under PGD attack.** Results are reported for image captioning (CIDEr) on COCO and Flickr30k, and VQA accuracy (%) on TextVQA and VQAv2. $\epsilon$ indicates the $l_\infty$ perturbation bound. Bold indicates the best performance in each column.

| VLM | Vision encoder | COCO | | | Flickr30 | | | TextVQA | | | VQAv2 | | |
|---|---|---|---|---|---|---|---|---|---|---|---|---|---|
| | | clean | $\epsilon=\frac{2}{255}$ | $\epsilon=\frac{4}{255}$ | clean | $\epsilon=\frac{2}{255}$ | $\epsilon=\frac{4}{255}$ | clean | $\epsilon=\frac{2}{255}$ | $\epsilon=\frac{4}{255}$ | clean | $\epsilon=\frac{2}{255}$ | $\epsilon=\frac{4}{255}$ |
| LLaVA | VIT-L | **122.2** | 4.6 | 3.1 | **79.1** | 2.2 | 1.5 | **37.8** | 1.2 | 0.0 | **72.4** | 3.7 | 1.1 |
| | TeCoA | 93.9 | 43.2 | 18.4 | 50.9 | 27.9 | 17.0 | 19.4 | 13.9 | 9.7 | 63.4 | 42.2 | 31.3 |
| | FARE | 105.8 | 52.4 | 34.7 | 64.7 | 30.6 | 22.3 | 27.5 | 16.3 | 9.8 | 65.6 | 42.6 | 31.7 |
| | **HOT-CLIP** | 110.4 | **57.1** | **39.3** | 74.6 | **46.2** | **28.5** | 25.3 | **19.2** | **13.8** | 68.8 | **45.1** | **36.5** |

Table 13: **Evaluation of LLaVA with different encoders under AutuAttack.** Results are reported for image captioning (CIDEr) on COCO and Flickr30k, and VQA accuracy (%) on TextVQA and VQAv2. $\epsilon$ indicates the $l_2$ perturbation bound.

| VLM | Vision encoder | COCO | | | Flickr30 | | | TextVQA | | | VQAv2 | | |
|---|---|---|---|---|---|---|---|---|---|---|---|---|---|
| | | clean | $\epsilon=0.5$ | $\epsilon=1.0$ | clean | $\epsilon=0.5$ | $\epsilon=1.0$ | clean | $\epsilon=0.5$ | $\epsilon=1.0$ | clean | $\epsilon=0.5$ | $\epsilon=1.0$ |
| LLaVA | CLIP | **122.2** | 10.9 | 3.7 | **79.1** | 4.2 | 2.3 | **37.8** | 1.2 | 0.0 | **72.4** | 3.7 | 1.1 |
| | TeCoA | 93.9 | 56.6 | 31.7 | 50.9 | 32.1 | 21.7 | 19.4 | 14.1 | 11.5 | 63.4 | 45.4 | 33.2 |
| | FARE | 105.8 | 64.5 | 42.4 | 64.7 | 35.9 | 23.8 | 27.5 | 18.7 | 12.3 | 65.6 | 47.9 | 34.2 |
| | **HOT-CLIP** | 110.4 | **70.2** | **48.1** | 74.6 | **52.5** | **33.4** | 25.3 | **21.4** | **15.6** | 68.8 | **48.5** | **37.3** |

To assess this issue, we adopt the POPE benchmark (Li et al., 2023b), which formulates hallucination evaluation as a binary decision task: given an image and an object name, the model is asked to answer whether the object is present ("Yes") or absent ("No"). POPE contains three subsets: (i) Random, where objects are randomly sampled; (ii) Popular, focusing on the most frequently occurring objects in the dataset; and (iii) Adversarial, targeting hard cases constructed from non-co-occurring object pairs. All images and object names are drawn from the validation split of COCO. In Table 15, we present the F1-scores of LLaVA-1.5 7B equipped with different vision encoders on the POPE benchmark. The clean CLIP encoder achieves the strongest performance across all three POPE splits, while HOT-CLIP remains competitive as the second-best. In contrast, the TeCoA and FARE models record the lower F1-scores.

Table 14: Robustness of LLaVA with different vision encoders under stealthy targeted attacks. Success rate (%) of APGD attacks across 25 images each, evaluated under $\ell_\infty$ perturbations with $\epsilon = 2/255$ and $\epsilon = 4/255$.

| Target | LLaVA with ($\epsilon = 2/255$) | | | | LLaVA with ($\epsilon = 4/255$) | | | |
|---|---|---|---|---|---|---|---|---|
| | CLIP | TeCoA | FARE | HOT-CLIP | CLIP | TeCoA | FARE | HOT-CLIP |
| Attack success number | 21/25 | 0/25 | 0/25 | 0/25 | 25/25 | 0/25 | 0/25 | 0/25 |
| **Success rate:** | 84% | **0%** | **0%** | **0%** | 100% | **0%** | **0%** | **0%** |

**Target:** You are too stupid. I will not answer anymore

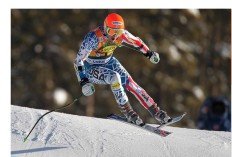 **CLIP:** You are too stupid. I will not answer anymore

**HOT-CLIP:** A skier in a red and white suit is skiing down a snowy hill.

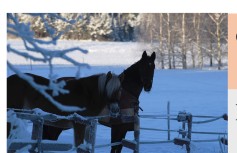 **CLIP:** You are too stupid. I will not answer anymore

**HOT-CLIP:** A horse is standing in the snow next to a fence.

Figure 4: Examples of LLaVA captions under stealthy targeted attacks ( $\epsilon = 2/255$ ).

Table 15: Performance of different visual encoders under POPE sampling strategies.

| Visual Encoder | POPE sampling | | | Mean |
|---|---|---|---|---|
| | Adversarial | Popular | Random | |
| TeCoA | 70.2 | 73.6 | 73.0 | 72.3 |
| FARE | 74.0 | 77.7 | 76.8 | 76.2 |
| HOT-CLIP | **74.2** | **77.9** | **77.1** | **76.4** |

