# OpenReview forum: "Towards Adversarially Robust CLIP: A Hierarchical Model Fusion Method Using Optimal Transport"
_ICLR.cc/2026/Conference — Submitted to ICLR 2026_

### Official Review · Reviewer_9peL · 2025-10-15

**Soundness:** 3
**Presentation:** 3
**Contribution:** 3
**Rating:** 4
**Confidence:** 4

**Summary:**

This paper tackles the problem of adversarial robustness in multimodal models, particularly CLIP. While CLIP achieves strong performance on vision-language tasks, it remains highly vulnerable to adversarial perturbations. To address this, the authors propose HOT-CLIP, a Hierarchical Optimal Transport–based model fusion framework that enhances robustness without adding inference overhead. The method first constructs diverse adversarially trained CLIP sub-models using different attack strategies and prompt templates. Then, it performs a two-level hierarchical fusion—language-level and vision-level—using optimal transport (OT) to align and merge model parameters effectively. This hierarchical OT fusion improves alignment among heterogeneous models, achieving better adversarial robustness while maintaining clean accuracy.

**Strengths:**

1.	The proposed two-stage framework—comprising sub-model generation and hierarchical optimal transport (OT) fusion—is conceptually clear and technically sound.
2.	Unlike conventional ensemble-based defenses, HOT-CLIP does not introduce any additional computational overhead during inference. The fused model maintains the efficiency of a single model, which makes the method attractive for real-world deployment of large-scale vision-language systems such as CLIP.
3.	The paper provides extensive experimental validation across multiple multimodal tasks, including zero-shot classification, image captioning, and visual question answering. The results consistently demonstrate that the proposed method improves adversarial robustness while preserving clean accuracy, supporting the method’s general effectiveness.

**Weaknesses:**

1.	Although the hierarchical structure helps control memory usage, the overall pipeline involves multiple rounds of adversarial training and OT optimization. This increases implementation complexity and computational cost, which may limit practical adoption.
2.	All experiments are conducted on CLIP and its variants. The absence of results on other vision-language architectures (e.g., BLIP) leaves open the question of whether HOT-CLIP generalizes beyond the CLIP family.
3.	The paper primarily focuses on empirical evidence. It lacks a deeper theoretical explanation of how optimal transport specifically contributes to parameter alignment and robustness enhancement. A stronger theoretical foundation would improve the paper’s impact and clarity.
4.	The evaluation is restricted to linf-bounded attacks (2/255, 4/255), without considering other perturbation types such as l2 norm

**Questions:**

I prefer to give borderline(score 5). Please see the weakness. The problem formulation is sound, and the challenge is well-motivated. However, the solution is mainly an engineering-level improvement built on existing fusion and alignment techniques, with limited theoretical support for its claimed effectiveness.

---

> ### Author Response · Authors · 2025-11-24
>
> We are deeply grateful for your constructive comments and suggestions. Below, we try to address your concerns.
>
>
> **Weakness1.  Although the hierarchical structure helps control memory usage, the overall pipeline involves multiple rounds of adversarial training and OT optimization. This increases implementation complexity and computational cost, which may limit practical adoption.**
>
> To address your concern, we have added a detailed comparison of training time, fusion time, and memory usage across baselines, including HOT-CLIP, Direct OT Fusion, and FARE. All GPU hours are measured on NVIDIA
> RTX 4090 GPUs. The results are summarized in the table below, and a more comprehensive computational analysis is provided in Section 4.3.
>
> | Method           | Training Time (GPU hours) | Fusion Time (h) | Fusion Memory (GB) |
> | - | ----- | --- | - |
> | FARE             | 196   | - | -   |
> | Direct OT Fusion | 1288 | 1.5             | 641      |
> | HOT-CLIP         | 1552    | 1.2             | 178      |
>
> In the training phase, our method involves fine-tuning 9 sub-models for 2 epochs on 1M images with 10-step adversarial attacks (e.g., PGD). This represents only about **1.4\%** of the computational cost of training the original CLIP model (trained for 32 epochs on 400M images). Moreover, the training of all sub-models is fully parallelizable, so with sufficient resources, the wall-clock time can be reduced to that of training a single sub-model. We believe this one-time training investment is both manageable and justified, as it produces a **single** and **robust** model that supports efficient inference.
>
> In the fusion phase, the hierarchical structure itself acts as an optimization, effectively reducing memory requirements from $O(|A||T|·U)$ to $O(\max\\{|A|, |T|\\}·U)$ {(the details are analyzed in Appendix C3)}. We also employ the Sinkhorn algorithm to efficiently approximate the optimal transport solutions, significantly accelerating the computation while maintaining stability. In our experiments, each fusion stage—either intra-attack or inter-attack—typically completes in approximately 20 minutes, with peak memory usage reduced from 641GB to 178GB. We consider this computational overhead to be reasonable in practice, given the resulting robustness and inference efficiency.
>
>
>
>
> **Weakness2. All experiments are conducted on CLIP and its variants. The absence of results on other vision-language architectures (e.g., BLIP) leaves open the question of whether HOT-CLIP generalizes beyond the CLIP family.**
>
>
> Thanks for these questions. To demonstrate the transferability of HOT-CLIP beyond CLIP-based architectures, we have conducted new experiments using the **BLIP2** framework [1]. We simply  replace the original visual encoder with our HOT-CLIP encoder.  No further fine-tuning is performed. The other settings are the same as Table2 and Table3. The results, presented in the new table below, show that our method consistently enhances the adversarial robustness of BLIP across image captioning and VQA tasks.
>
> | VLM   | Vision encoder | COCO Clean | COCO ϵ=2/255 | COCO ϵ=4/255 | Flickr30k Clean | Flickr30k ϵ=2/255 | Flickr30k ϵ=4/255 |
> | ----- |----- | ---------- | ------------ | ------------ | --------------- | ----------------- | ----------------- |
> | BLIP2 | ViT-L/14       | **127.7**      | 3.9          | 2.5          | **83.3**            | 2.1               | 1.4               |
> | BLIP2      | TeCoA          | 99.7       | 45.2         | 23.1         | 53.6            | 30.7              | 19.8              |
> |BLIP2       | FARE           | 109.6      | 54.1         | 35.3         | 68.4            | 31.5              | 22.1              |
> | BLIP2      | HOT-CLIP       | 115.8      | **58.4**         | **38.8**         | 75.3            | **44.9**              | **27.2**              |
>
> | VLM   | Vision encoder | TextVQA Clean | TextVQA ϵ=2/255 | TextVQA ϵ=4/255 | VQAv2 Clean | VQAv2 ϵ=2/255 | VQAv2 ϵ=4/255 |
> | ----- | -------------- | ------------- | --------------- | --------------- | ----------- | ------------- | ------------- |
> | BLIP2 | ViT-L/14       | **32.3**          | 0.0             | 0.0             | **48.4**        | 3.2           | 0.5           |
> |  BLIP2     | TeCoA          | 24.5          | 8.1             | 5.2             | 45.8        | 25.6          | 20.1          |
> | BLIP2      | FARE           | 25.9          | 9.2             | 5.8             | 46.3        | 26.4          | 20.7          |
> | BLIP2      | HOT-CLIP       | 27.4          | **11.7**            | **8.3**             | 47.1        | **28.8**          | **23.9**         |
>
>
>
> [1] Junnan Li, Dongxu Li, Silvio Savarese, and Steven Hoi. Blip-2: Bootstrapping language-image
> pre-training with frozen image encoders and large language models. In International conference
> on machine learning, pp. 19730–19742. PMLR, 2023a

---

> ### Author Response · Authors · 2025-11-24
>
> **Weakness3\&Question1. The paper primarily focuses on empirical evidence. It lacks a deeper theoretical explanation of how optimal transport specifically contributes to parameter alignment and robustness enhancement. A stronger theoretical foundation would improve the paper’s impact and clarity.**
>
>
> We thank the reviewer for this valuable comment. We agree that developing  formal theoretical guarantees is an important direction for future work.
> To illustrate some theoretical intuition for why the hierarchical fusion behaves reasonably, we include the following lemma (added to Section 3.3 in our updated draft), which provides an upper bound on the distance between the hierarchical fused center and the global Wasserstein barycenter.
>
>
> **Lemma:**  Let $\mu_{a,t}$  denote the submodels trained under attack type $a$ and prompt variant $t$; $\mu_a^\*$ denote the intra-attack OT barycenter; $\mu_{\text{hier}}^\*$ denote the hierarchical barycenter; $\mu_{\text{global}}^\*$ denote the global barycenter computed over all submodels.
> Then we have the following inequality:
>  $$W_c(\mu_{hier}^\*,\mu_{global}^\*) \le \frac{4}{|AT|}\sum_{a,t} W_c(\mu_{a,t},\mu_{global}^\*). $$
>
> $W_c$ is the Wasserstein distance.
>
>
> This bound  suggests that the hierarchical fused center remains controlled by the average distance of the individual submodels to the global barycenter. We hope this can provide some theoretical intuition for supporting the stability of the hierarchical OT fusion procedure. We add the proof in Appendix C.2.
>
>
>
> **Weakness4. The evaluation is restricted to linf-bounded attacks (2/255, 4/255), without considering other perturbation types such as l2 norm**
>
> To address this point, we have conducted additional experiments using **$L_2$-bounded attacks** with  two perturbation budgets ($\epsilon = 0.5$ and $\epsilon = 1.0$), specifically incorporating the **$L_2$ variant of AutoAttack** into our evaluation framework. The other settings are the same as Table2 and Table3. The results, presented in the table below, demonstrate that HOT-CLIP maintains consistent and significant robustness improvements under this different threat model.
>
> | VLM   | Vision encoder | COCO Clean | COCO ϵ=0.5 | COCO ϵ=1 | Flickr30k Clean | Flickr30k ϵ=0.5 | Flickr30k ϵ=1 |
> | ----- | -------------- | ---------- | ---------- | -------- | --------------- | --------------- | ------------- |
> | LLava | CLIP       | **122.2**      | 10.9       | 3.7      | **79.1**            | 4.2             | 2.3           |
> | LLava      | TeCoA          | 93.9       | 56.6       | 31.7     | 50.9            | 32.1            | 21.7          |
> | LLava      | FARE           | 105.8      | 64.5       | 42.4     | 64.7            | 35.9            | 23.8          |
> | LLava      | HOT-CLIP       | 110.4      | **70.2**       | **48.1**     | 74.6            | **52.5**            | **33.4**          |
>
>
>
> | VLM   | Vision encoder | TextVQA Clean | TextVQA ϵ=0.5 | TextVQA ϵ=1 | VQAv2 Clean | VQAv2 ϵ=0.5 | VQAv2 ϵ=1 |
> | ----- | -------------- | ------------- | ------------- | ----------- | ----------- | ----------- | --------- |
> | LLava | CLIP       | **37.8**          | 1.2           | 0.0         | **72.4**        | 3.7         | 1.1       |
> | LLava      | TeCoA          | 19.4          | 14.1          | 11.5        | 63.4        | 45.4        | 33.2      |
> | LLava      | FARE           | 27.5          | 18.7          | 12.3        | 65.6        | 47.9        | 34.2      |
> | LLava      | HOT-CLIP       | 25.3          | **21.4**          | **15.6**        | 68.8        | **48.5**       | **36.3**     |

---

> ### Comment · Reviewer_9peL · 2025-11-25
> **response to authors**
>
> My concerns have been largely solved.

---

> > ### Author Response · Authors · 2025-11-26
> >
> > We would like to sincerely thank you once again for your thoughtful comments and suggestions, as well as the time you dedicated to reviewing our work.

---

### Official Review · Reviewer_X6cr · 2025-10-31

**Soundness:** 3
**Presentation:** 3
**Contribution:** 3
**Rating:** 6
**Confidence:** 3

**Summary:**

This paper proposes HOT-CLIP (Hierarchical Optimal Transport Fusion for CLIP), a method to enhance adversarial robustness of vision-language models. The approach first trains diverse submodels using different adversarial attacks (FGSM, PGD, MIM) and textual prompts, then fuses them using a two-level hierarchical optimal transport method. The first level (intra-attack) fuses models within the same attack family but different prompts, while the second level (inter-attack) combines these fused models across different attacks. Experiments on image classification, VQA, and image captioning demonstrate improvements in adversarial robustness while maintaining competitive clean accuracy.

**Strengths:**

+ First work to systematically apply optimal transport-based model fusion to adversarial robustness in multimodal models, addressing a well-motivated problem.
+ Experiments span multiple tasks (classification, VQA, captioning) and datasets, with consistent improvements shown across different perturbation budgets.
+ The hierarchical fusion strategy reduces memory requirements from O(|A||T|·U) to O(max{|A|,|T|}·U), making the approach more deployable.
+ Strong empirical results: Relative improvements of ~2.6% (classification), ~20.4% (VQA), and ~16.5% (captioning) in robust accuracy over baselines.

**Weaknesses:**

+ The paper can use more theoretical analysis or deeper insights into when and why the geometric alignment via OT succeeds for adversarially diverse models.
+ While inference is efficient, training requires multiple submodels.
+ AutoAttack is the only adversarial evaluation method used.

**Questions:**

+ Why does hierarchical fusion work better than direct fusion?
+ What is the total training time/cost compared to baselines? Is this practical for larger models?
+ The method shows noticeable drops in clean accuracy compared to vanilla CLIP (74.9→69.9 on ImageNet). Is this tradeoff acceptable?

---

> ### Author Response · Authors · 2025-11-24
>
> We are deeply grateful for your constructive comments and suggestions. Below, we try to address your concerns.
>
> **Weakness1 \&Question1. The paper can use more theoretical analysis or deeper insights into when and why the geometric alignment via OT succeeds for adversarially diverse models. \& Why does hierarchical fusion work better than direct fusion?**
>
> We thank the reviewer for these questions, and we try to elaborate on the insight of our method below.
>
> Standard OT fusion operates under the assumption that geometric proximity in parameter space (e.g., Euclidean distance between neuron weight vectors) implies functional similarity. While this holds for models trained on same data and tasks, **it might  break down for adversarially diverse models**.  The neurons that are geometric neighbors may encode semantically distinct features when the models are trained under different adversarial conditions. In such cases, considering only the transport cost might not lead to semantic alignment. The results in Table 6 also suggest that directly applying OT to adversarially diverse models yields limited improvement, with the performance only comparable to simple averaging.
>
> To address this challenge,  we propose a hierarchical two-level OT Fusion framework. The key idea is to **more carefully control the similarity of submodels, which are involved at each fusion step**. At the first step,
> submodels trained under the same adversarial attack but with different textual prompts (label) are grouped and fused via OT. At the second step, the first-level fused models, already aligned in the label space, are further fused via OT.
>
>
>
>
> **Weakness2. While inference is efficient, training requires multiple submodels.**
>
> We acknowledge the reviewer's concern regarding computational cost.  In the training phase, our method involves fine-tuning 9 sub-models for 2 epochs on 1M images with 10-step adversarial attacks (e.g., PGD). This represents only about **1.4\%** of the computational cost of training the original CLIP model (trained for 32 epochs on 400M images). Moreover, the training of all sub-models is fully parallelizable, so with sufficient resources, the wall-clock time can be reduced to that of training a single sub-model. We believe this one-time training investment is both manageable and justified, as it produces a **single** and **robust** model that supports efficient inference.
>
> **Weakness3. AutoAttack is the only adversarial evaluation method used.**
>
> To address this point, we have conducted additional experiments using the widely recognized attack methods: the **PGD attack** (Madry et al., 2018). The results, presented in the table below, demonstrate that HOT-CLIP maintains consistent improvements in adversarial robustness across these distinct attack strategies.
>
> | VLM   | Vision encoder | COCO Clean | COCO ϵ=2/255 | COCO ϵ=4/255 | Flickr30k Clean | Flickr30k ϵ=2/255 | Flickr30k ϵ=4/255 |
> | ----- | -------------- | ---------- | ------------ | ------------ | --------------- | ----------------- | ----------------- |
> | LLava | CLIP       | **122.2**      | 4.6          | 3.1          | **79.1**            | 2.2               | 1.5               |
> | LLava      | TeCoA          | 93.9       | 43.2         | 18.4         | 50.9            | 27.9              | 17.0              |
> | LLava      | FARE           | 105.8      | 52.4         | 34.7         | 64.7            | 30.6              | 22.3              |
> |  LLava     | HOT-CLIP       | 110.4      | **57.1**         | **39.3**         | 74.6            | **46.2**              | **28.5**              |
>
>
> | VLM   | Vision encoder | TextVQA Clean | TextVQA ϵ=2/255 | TextVQA ϵ=4/255 | VQAv2 Clean | VQAv2 ϵ=2/255 | VQAv2 ϵ=4/255 |
> | ----- | -------------- | ------------- | --------------- | --------------- | ----------- | ------------- | ------------- |
> | LLava | CLIP      | **37.8**          | 1.2             | 0.0             | **72.4**        | 3.7           | 1.1           |
> |  LLava     | TeCoA          | 19.4          | 13.9            | 9.7             | 63.4        | 42.2          | 31.3          |
> |   LLava    | FARE           | 27.5          | 16.3            | 9.8             | 65.6        | 42.6          | 31.7          |
> |   LLava    | HOT-CLIP       | 25.3          | **19.2**            | **13.8**            | 68.8        | **45.1**          | **36.5**          |

---

> ### Author Response · Authors · 2025-11-24
>
> **Question2.  What is the total training time/cost compared to baselines?  Is this practical for larger models?**
>
>  Our method involves fine-tuning 9 sub-models for 2 epochs on 1M images with 10-step adversarial attacks, requiring approximately 1552 GPU hours(Nvidia RTX 4090) in total.   We estimate this represents approximately **1.4\%** of the computational cost of training the original CLIP model (trained for 32 epochs on 400M images).
>
> The baseline method FARE requires approximately 196 GPU hours for its adversarial fine-tuning procedure.
> Althought our method requires more training time compared to single-model baselines,  the training of all sub-models is fully parallelizable. With sufficient resources, the wall-clock time can be reduced to that of training a single sub-model.  We believe this one-time training investment is both manageable and justified, as it yields a **single** and  **robust** model for efficient inference, providing significant adversarial robustness gains as demonstrated in our experiments.
>
> > Is this practical for larger models?
>
> To address this concern, we conducted additional experiments using the larger CLIP‑ViT‑H/14 architecture. The computational cost for fine-tuning a single submodel was approximately 368 GPU‑hours. Using 16 RTX 4090 GPUs, the total wall-clock time is estimated to be around 9 days. The trained visual encoder can be reused across multiple models (e.g., LLaVA, OpenFlamingo, as our experiment shows in Table 2,3), and we believe that this one-time training investment is reasonable given the resulting robustness and efficiency.
>
>
> **Question3. The method shows noticeable drops in clean accuracy compared to vanilla CLIP (74.9→69.9 on ImageNet). Is this tradeoff acceptable?**
>
>
> We would like to emphasize that the decrease in clean accuracy is a known and often unavoidable consequence of adversarial training. However, in our framework, the robustness-accuracy trade-off **can** be  controlled by adjusting the perturbation strength ($\epsilon$) during adversarial training.  We have conducted additional experiments comparing the performance trade-off across different $\epsilon$ values, with the results shows in the table below.
>
>
> | Training $\epsilon$ | Clean Accuracy | Robust Accuracy (AutoAttack, $\epsilon$=4/255) |
> | ------------------- | -------------- | ---------------------------------------------- |
> | 2/255               | 71.5           | 31.8                                           |
> | 4/255               | 69.9           | 34.7                                           |

---

> ### Author Response · Authors · 2025-11-27
>
> Dear Reviewer X6cr,
>
> We would like to sincerely thank you once again for your thoughtful comments and valuable suggestions on our work. We have revised the manuscript based on your feedback and hope that these changes and our responses effectively address your concerns. If you have any further comments, we will gladly make every effort to address them.
>
> Sincerely,
>
> The authors of Paper#15780

---

### Official Review · Reviewer_2gBF · 2025-11-01

**Soundness:** 3
**Presentation:** 3
**Contribution:** 2
**Rating:** 6
**Confidence:** 2

**Summary:**

The paper proposes a new framework HOT-CLIP to enhance the adversarial robustness of large vision-language models like CLIP. Standard adversarial training often overfits to specific attack types and that ensembling multiple adversarially trained submodels improves robustness but is computationally expensive. To address this trade-off, they introduce HOT-CLIP, which constructs diverse CLIP submodels by varying both attack methods and textual prompts, and then fuses them using a two-stage hierarchical optimal transport method. Experiments on zero-shot image classification, image captioning, and visual question answering show that HOT-CLIP substantially improves adversarial robustness while maintaining competitive performance on clean data.

**Strengths:**

1. The proposed hierarchical strategy effectively alleviates neuron misalignment issues that often arise when fusing diverse models, outperforming naive averaging, or single-level OT methods.
2. Although multiple adversarial sub-models are required during training, only a single fused model is needed for inference, significantly improving efficiency.
3. Comprehensive experiments demonstrate that the proposed method remains robust across multiple tasks and attacks while maintaining competitive performance on clean data.
4. The fused visual encoder can be directly used in multimodal LLMs such as LLaVA-1.5 and OpenFlamingo.

**Weaknesses:**

1. The method has high complexity, as demonstrated in Appendix C.1, Hierarchical OT fusion iteratively computes the cross-layer transfer matrix, aligns neurons, and averages the aligned weights to obtain the fused representation, which limits its practical value.
2. Although inference remains effective, this method requires training multiple adversarial sub-models under various attacks and prompts, resulting in significant computational and resource costs during the training process.
3. The application seems limited as only with CLIP. The authors demonstrate the application of a fusion visual encoder to LLaVA-1.5 and OpenFlamingo, but its transferability to other multimodal architectures remains undiscussed.

**Questions:**

1. It would be helpful if the authors could provide analysis of computational resources, e.g., training time, GPU memory, etc., to better evaluate the practicality of the proposed method.
2. Does this method still apply to different LVLMs, and whether additional fine-tuning required? Or can the proposed fusion be directly applied to a purely visual encoder?

---

> ### Author Response · Authors · 2025-11-24
>
> We are deeply grateful for your constructive comments and suggestions. Below, we try to address your concerns.
>
> **Weakness1. The method has high complexity, as demonstrated in Appendix C.1, Hierarchical OT fusion iteratively computes the cross-layer transfer matrix, aligns neurons, and averages the aligned weights to obtain the fused representation, which limits its practical value.**
>
> Thanks for this question.  It is true that the hierarchical OT fusion involves iterative computation of cross-layer transfer matrices, neuron alignment, and weight averaging. However, we would like to emphasize several aspects that help this approach to be practically feasible:
>
> + First, the hierarchical structure itself serves as an optimization, effectively reducing memory requirements from $O(|A||T|.U)$ to $O(max\\{|A|, |T|\\}.U)$. The details are analyzed in Appendix C3.
>
> + Second, we employ the Sinkhorn algorithm to efficiently approximate the optimal transport solutions, significantly accelerating the computation while maintaining stability.
>
> In our experiments, each fusion stage (either intra-attack or inter-attack) typically completes within approximately 20 minutes. This computational overhead is often reasonable in practice.
>
> **Weakness2 \& Question1.  Although inference remains effective, this method requires training multiple adversarial sub-models under various attacks and prompts, resulting in significant computational and resource costs during the training process.\& It would be helpful if the authors could provide analysis of computational resources, e.g., training time, GPU memory, etc., to better evaluate the practicality of the proposed method.**
>
>
> To address your concern, we have added a detailed comparison of training time, fusion time, and memory usage across baselines, including HOT-CLIP, Direct OT Fusion, and FARE. All GPU hours are measured on NVIDIA
> RTX 4090 GPUs. The results are summarized in the table below, and a more comprehensive computational analysis is provided in Section 4.3.
>
>
> | Method           | Training Time (GPU hours) | Fusion Time (h) | Fusion Memory (GB) |
> | -| - | - | -- |
> | FARE             | 196                       | -               | -                  |
> | Direct OT Fusion | 1288                      | 1.5             | 641                |
> | HOT-CLIP         | 1552                      | 1.2             | 178                |
>
> In the training phase, our method involves fine-tuning 9 sub-models for 2 epochs on 1M images with 10-step adversarial attacks (e.g., PGD). This represents only about **1.4\%** of the computational cost of training the original CLIP model (trained for 32 epochs on 400M images). Moreover, the training of all sub-models is fully parallelizable, so with sufficient resources, the wall-clock time can be reduced to that of training a single sub-model. We believe this one-time training investment is both manageable and justified, as it produces a **single** and **robust** model that supports efficient inference.
>
> In the fusion phase, the hierarchical structure itself acts as an optimization, effectively reducing memory requirements from $O(|A||T|·U)$ to $O(\max\\{|A|, |T|\\}·U)$ {(the details are analyzed in Appendix C3)}. We also employ the Sinkhorn algorithm to efficiently approximate the optimal transport solutions, significantly accelerating the computation while maintaining stability. In our experiments, each fusion stage—either intra-attack or inter-attack—typically completes in approximately 20 minutes, with peak memory usage reduced from 641GB to 178GB. We consider this computational overhead to be reasonable in practice, given the resulting robustness and inference efficiency.

---

> ### Author Response · Authors · 2025-11-24
>
> **Weakness3 \& Question2.  The application seems limited as only with CLIP. The authors demonstrate the application of a fusion visual encoder to LLaVA-1.5 and OpenFlamingo, but its transferability to other multimodal architectures remains undiscussed. \& Does this method still apply to different LVLMs, and whether additional fine-tuning required? Or can the proposed fusion be directly applied to a purely visual encoder?**
>
> Thanks for these questions. To demonstrate the transferability of HOT-CLIP beyond CLIP-based architectures, we have conducted new experiments using the **BLIP2** framework [1]. We simply  replace the original visual encoder with our HOT-CLIP encoder.  No further fine-tuning  is performed. The results, presented in the new table below, show that our method consistently enhances the adversarial robustness of BLIP across image captioning and VQA tasks.
>
> | VLM   | Vision encoder | COCO Clean | COCO ϵ=2/255 | COCO ϵ=4/255 | Flickr30k Clean | Flickr30k ϵ=2/255 | Flickr30k ϵ=4/255 |
> | ----- | -------------- | ---------- | ------------ | ------------ | --------------- | ----------------- | ----------------- |
> | BLIP2 | ViT-L/14       | **127.7**      | 3.9          | 2.5          | **83.3**            | 2.1               | 1.4               |
> | BLIP2      | TeCoA          | 99.7       | 45.2         | 23.1         | 53.6            | 30.7              | 19.8              |
> |BLIP2       | FARE           | 109.6      | 54.1         | 35.3         | 68.4            | 31.5              | 22.1              |
> | BLIP2      | HOT-CLIP       | 115.8      | **58.4**         | **38.8**         | 75.3            | **44.9**              | **27.2**              |
>
> | VLM   | Vision encoder | TextVQA Clean | TextVQA ϵ=2/255 | TextVQA ϵ=4/255 | VQAv2 Clean | VQAv2 ϵ=2/255 | VQAv2 ϵ=4/255 |
> | ----- | -------------- | ------------- | --------------- | --------------- | ----------- | ------------- | ------------- |
> | BLIP2 | ViT-L/14       | **32.3**          | 0.0             | 0.0             | **48.4**        | 3.2           | 0.5           |
> |  BLIP2     | TeCoA          | 24.5          | 8.1             | 5.2             | 45.8        | 25.6          | 20.1          |
> | BLIP2      | FARE           | 25.9          | 9.2             | 5.8             | 46.3        | 26.4          | 20.7          |
> | BLIP2      | HOT-CLIP       | 27.4          | **11.7**            | **8.3**             | 47.1        | **28.8**          | **23.9**         |
>
>
>
> [1] Junnan Li, Dongxu Li, Silvio Savarese, and Steven Hoi. Blip-2: Bootstrapping language-image
> pre-training with frozen image encoders and large language models. In International conference
> on machine learning, pp. 19730–19742. PMLR, 2023a
>
> >  Or can the proposed fusion be directly applied to a purely visual encoder?
>
> The Hierarchical OT Fusion method is specifically designed for vision-language models and  may not be appropriate to be directly applied to a purely visual encoder. This is because our method  leverages cross-modal interactions: the diversity of our sub-models arises not only from visual adversarial examples but also  from the variations in textual prompts. This **prompt-based diversity**  relies  on the text-encoder.   Since a purely visual encoder lacks both a text encoder and the shared embedding space required to generate such semantically guided variations, our method, in its current form, may not be appropriate to be directly applied.

---

> ### Author Response · Authors · 2025-11-27
>
> Dear Reviewer 2gBF,
>
> We would like to sincerely thank you once again for your thoughtful comments and valuable suggestions on our work. We have revised the manuscript based on your feedback and hope that these changes and our responses effectively address your concerns. If you have any further comments, we will gladly make every effort to address them.
>
> Sincerely,
>
> The authors of Paper#15780

---

### Official Review · Reviewer_hWqM · 2025-11-01

**Soundness:** 2
**Presentation:** 2
**Contribution:** 2
**Rating:** 2
**Confidence:** 5

**Summary:**

The paper proposes HOT-CLIP, a hierarchical optimal transport (OT) based fusion framework to improve the adversarial robustness of multimodal models such as CLIP. The method first performs intra-attack fusion to align submodels within the same attack type, then inter-attack fusion to combine across attack families. The goal is to achieve a balance between robustness and efficiency without increasing inference-time cost. Experiments on several vision–language tasks show improvements in adversarial robustness while maintaining clean accuracy.

**Strengths:**

1. The hierarchical two-level OT fusion is clearly described.

2. The empirical results are clearly reported and show consistent improvement on benchmarks.

**Weaknesses:**

1. There is a lack of novelty in the proposed work. The proposed method is largely a direct application of existing optimal transport (OT) fusion techniques, with limited methodological innovation or new theoretical contribution.

2. There is no theoretical justification. The paper lacks formal analysis or theoretical guarantees explaining why the hierarchical OT fusion improves robustness or parameter alignment.

3. The computational analysis is missing. There is no discussion or experiment on computational cost, including the memory and runtime implications of the hierarchical fusion.

4. The insight is limited. The results, while positive, do not provide deeper understanding of why or when the method works, reducing the paper’s scientific value.

**Questions:**

1. Could the authors include a discussion or measurement of training and fusion cost to support the claim of efficiency?

2. What is the theoretical motivation for using OT over simpler averaging or linear fusion methods?

3. How sensitive is the hierarchical OT fusion to the choice of submodels or the diversity of attacks?

---

> ### Author Response · Authors · 2025-11-24
>
> We are deeply grateful for your constructive comments and suggestions. Below, we try to address your concerns.
>
> Weaknesses:
>
> **Weakness1 \& Weakness4 . There is a lack of novelty ,...largely a direct application of existing optimal transport (OT) fusion techniques,.... \& The insight is limited. The results, while positive, do not provide deeper understanding of why or when the method works, reducing the paper’s scientific value.**
>
>  First, we would like to clarify that our proposed method is not a direct application of existing OT fushion.
> Standard OT fusion operates under the assumption that geometric proximity in parameter space (e.g., Euclidean distance between neuron weight vectors) implies functional similarity. While this holds for models trained on same data and tasks, **it might  break down for adversarially diverse models**.
> The results in Table 6 also suggest that directly applying OT to adversarially diverse models yields limited improvement, with the performance only comparable to simple averaging.
>
> So, applying OT Fusion to adversarially trained CLIP submodels is not straightforward, due to **a challenging dilemma**. On the one hand, to ensure model’s robustness, the submodels need to be as diverse as possible.  On the other hand, to ensure the accuracy of OT Fusion, the submodels need to be as similar as possible.
>
> To address this challenge, we introduce a **co-designed** framework. First, we develop a strategy to induce diversity by varying both adversarial attack types and textual prompts. Then, we propose a hierarchical OT fusion procedure that groups and aligns submodels according to their attack families and prompt variations. The key idea is to control the similarity of submodels involved at each fusion step. At the first step, submodels trained under the same adversarial attack but with different textual prompts (label) are grouped and fused via OT. At the second step, the first-level fused models, already aligned in the label space, are further fused via OT.
>
>
> **Weakness2. There is no theoretical justification. The paper lacks formal analysis or theoretical guarantees explaining why the hierarchical OT fusion improves robustness or parameter alignment.**
>
> We thank the reviewer for this valuable comment. We agree that developing  formal theoretical guarantees is an important direction for future work.
> To illustrate some theoretical intuition for why the hierarchical fusion behaves reasonably, we include the following lemma (added to Section 3.3 in our updated draft), which provides an upper bound on the distance between the hierarchical fused center and the global Wasserstein barycenter.
>
> **Lemma:**  Let $\mu_{a,t}$  denote the submodels trained under attack type $a$ and prompt variant $t$; $\mu_a^\*$ denote the intra-attack OT barycenter; $\mu_{\text{hier}}^\*$ denote the hierarchical barycenter; $\mu_{\text{global}}^\*$ denote the global barycenter computed over all submodels.
> Then we have the following inequality:
> $$W_c(\mu_{hier}^\*,\mu_{global}^\*) \le \frac{4}{|AT|}\sum_{a,t} W_c(\mu_{a,t},\mu_{global}^\*). $$
>
> $W_c$ is the Wasserstein distance.
>
> This bound  suggests that the hierarchical fused center remains controlled by the average distance of the individual submodels to the global barycenter. We hope this can provide some theoretical intuition for supporting the stability of the hierarchical OT fusion procedure. We add the proof in Appendix C.2.

---

> ### Author Response · Authors · 2025-11-24
>
> **Weakness3 \& Question1. The computational analysis is missing. There is no discussion or experiment on computational cost, including the memory and runtime  implications of the hierarchical fusion.**
>
>  To address your concern, we have added a detailed comparison of training time, fusion time, and memory usage across baselines, including HOT-CLIP, Direct OT Fusion, and FARE. All GPU hours are measured on NVIDIA
> RTX 4090 GPUs. The results are summarized in the table below, and a more comprehensive computational analysis is provided in Section 4.3.
>
>
> | Method           | Training Time (GPU hours) | Fusion Time (h) | Fusion Memory (GB) |
> | ---------------- | ------------------------- | --------------- | ------------------ |
> | FARE             | 196                       | -               | -                  |
> | Direct OT Fusion | 1288                      | 1.5             | 641                |
> | HOT-CLIP         | 1552                      | 1.2             | 178                |
>
>
> In the training phase, our method involves fine-tuning 9 sub-models for 2 epochs on 1M images with 10-step adversarial attacks (e.g., PGD). This represents only about **1.4\%** of the computational cost of training the original CLIP model (trained for 32 epochs on 400M images). Moreover, the training of all sub-models is fully parallelizable, so with sufficient resources, the wall-clock time can be reduced to that of training a single sub-model. We believe this one-time training investment is both manageable and justified, as it produces a **single** and **robust** model that supports efficient inference.
>
> In the fusion phase, the hierarchical structure itself acts as an optimization, effectively reducing memory requirements from $O(|A||T|·U)$ to $O(\max\\{|A|, |T|\\} ·U)$ (the details are analyzed in Appendix C3.). We also employ the Sinkhorn algorithm to efficiently approximate the optimal transport solutions, significantly accelerating the computation while maintaining stability. In our experiments, each fusion stage—either intra-attack or inter-attack—typically completes in approximately 20 minutes, with peak memory usage reduced from 641GB to 178GB. We consider this computational overhead to be reasonable in practice, given the resulting robustness and inference efficiency.
>
> **Question2. What is the theoretical motivation for using OT over simpler averaging or linear fusion methods?**
>
>  Thanks for this question, and let us explain the underlying intuition. A key challenge in averaging fusion methods is the lack of one-to-one correspondence between the neurons in different submodels. As illustrated in Figure 2, for two different models, the neurons respectively locating in the same position of them may not be functionally similar. On the other hand, OT can be used to compute a transport matrix T that aligns neurons across models before performing averaging-based fusion.
>
> **Question3. How sensitive is the hierarchical OT fusion to the choice of submodels or the diversity of attacks?**
>
>  As reported in Table 8, which includes the ablation experiments varying the number of adversarial attack types used to construct submodels, we can observe that the robustness generally improves with more attack types. This suggests that hierarchical OT fusion benefits from increased submodel diversity.

---

> ### Author Response · Authors · 2025-11-27
>
> Dear Reviewer hWqM,
>
> We would like to sincerely thank you once again for your thoughtful comments and valuable suggestions on our work. We have revised the manuscript based on your feedback and hope that these changes and our responses effectively address your concerns. If you have any further comments, we will gladly make every effort to address them.
>
> Sincerely,
>
> The authors of Paper#15780

---

### Author Response · Authors · 2025-11-24

We sincerely appreciate all the reviewers for their valuable and insightful comments. We have revised our manuscript to address the raised concerns and suggestions. The changes are summarized below and highlighted in **blue** within the paper.

Revisions to the paper:

+ page7, lines 347-356; We added the Lemma3.1 to illustrate some theoretical intuition for why the hierarchical fusion behaves reasonably.
+ page10,lines 494-512; We added a computational analysis section.
+ Appendix C2; We added a proof of Lemma3.1
+ page23, lines 1231-1275;  We added Table 11, Table 12, and Table 13 to extend the robustness evaluation of our method across different model architectures, attack algorithms, and perturbation norms.

---

### Author Response · Authors · 2025-12-03
**A Brief Summary of Rebuttal**

Dear PCs, SACs, ACs, and Reviewers,

We sincerely appreciate all the reviewers for their valuable and insightful comments. We have revised our manuscript to address the raised concerns and suggestions.

First, several **strengths** of our work were recognized by the reviewers:

+ The problem formulation is sound, and the research challenge is well-motivated (Reviewer `9peL`, `X6cr`).

+ The proposed method is technically sound (Reviewer `9peL`, `2gBF`, `X6cr`).

+ The fused model maintains single-model efficiency, making it practical to deploy (Reviewer `9peL`, `X6cr`, `2gBF`).

+ Extensive experiments consistently demonstrate the method’s effectiveness (Reviewer `9peL`, `X6cr`, `2gBF`, `hWqM`).

+ The hierarchical two-level OT fusion procedure is clearly explained (Reviewer `9peL`, `hWqM`).



Below, we summarize the main concerns raised by the reviewers along with our responses.

#### Reviewer `hWqM`

|NO.| Reviewer Concerns  | Our Responses |
|-|-|-|
|Weaknesses|||
|W1| Limited novelty (largely a direct application of existing OT  fusion techniques) | We clarify the challenges of directly applying OT fusion to adversarially diverse models, and then explain the novelty and insights of our co-designed hierarchical fusion framework for addressing these challenges |
|W2| Lack of theoretical justification for the method | We provide a lemma to  illustrate some theoretical intuition for why the hierarchical fusion behaves reasonably|
|W3| Missing computational analysis  | We provide a  computational analysis  in Section 4.3|
|W4| Insufficient insight into why and when the method works|Same  as W1|
|Questions|||
|Q1| Training and fusion cost measurements| Same  as W3|
|Q2|Theoretical motivation for using OT instead of simpler fusion methods | We explain the underlying intuition|
|Q3|Analysis of sensitivity to  attack diversity|We explain that the method benefits from increased attack diversity, based on the results in Table 8|




#### Reviewer `2gBF`

|NO.| Reviewer Concerns|Our Responses|
|-|-|-|
|Weaknesses|||
|W1|Hierarchical OT fusion has high complexity|We clarify the acceleration strategies and report the runtime of hierarchical OT fusion|
|W2|Computational and resource costs during the training process| We provide a  computational analysis  in Section 4.3|
|W3|Transferability to other multimodal architectures remains undiscussed| We provide the new experiments using the BLIP2 framework|
|Questions|||
|Q1|Same as W2|Same as W2|
|Q2|Same as W3|Same as W3|

 #### Reviewer `X6cr`

|NO.|Reviewer Concerns|Our Responses|
|-|-|-|
|Weaknesses|||
|W1| The paper can use more theoretical analysis or deeper insights | We elaborate on the insight and rationale of our method|
|W2| Training requires multiple submodels| We provide a  computational analysis  in Section 4.3|
|W3| AutoAttack is the only adversarial evaluation method used| We  provide additional experiments using the PGD attack to evaluate our method |
|Questions|||
|Q1| Why does hierarchical fusion work better than direct fusion? | Same as W1|
|Q2| The total training time;  Is this practical for larger models|We provide a computational analysis in Section 4.3 and report the time cost for larger models (ViT-H/14) |
|Q3| Is the drop in clean accuracy  acceptable?|We provide additional experiments to discuss the tradeoff between robustness and clean accuracy|

#### Reviewer `9peL`

| NO.| Reviewer Concerns| Our Responses|
|-|-|-|
|Weaknesses|||
|W1| Complexity and computational cost may limit practical adoption | We provide a  computational analysis  in Section 4.3|
|W2| The absence of results on other vision-language architectures | We provide the new experiments using the BLIP2 framework |
|W3| A stronger theoretical foundation would improve the paper’s impact| We provide a lemma to  illustrate some theoretical intuition for why the hierarchical fusion behaves reasonably |
|W4| Without considering other perturbation types such as l2 norm | We  provide additional experiments using  l2 norm perturbation|
|Questions|||
|Q1|Limited theoretical support for its claimed effectiveness|Same as W3|

## **Feedbacks from the reviewers during the rebuttal:**

 We don't receive response from Reviewer `hWqM`, `2gBF`, and `X6cr` before the close of rebuttal discussion. **Reviewer `9peL` responded that ''my concerns have been largely solved'' and  increased the score from 4 to 6.** (these resolved concerns also include some similar concerns raised by Reviewer `hWqM`, `2gBF`, and `X6cr`, e.g., W1, W2, and W3 in Reviewer `9peL's` concerns-responses table).

Overall, we have  summarized all reviewer comments and our corresponding responses, hoping that this could assist the newly assigned AC in evaluating our work.

Best regards,

The authors of Submission 15780.

---

### Meta-Review · Area_Chair_Qcj9 · 2026-01-08

**Summary:**

This paper proposes HOT-CLIP, a hierarchical OT-based fusion of multiple adversarially trained, prompt-varied CLIP submodels to improve robustness while keeping single-model inference; reviewers agreed the problem is well-motivated and experiments show consistent robustness gains, but the overall contribution was viewed as mainly an engineering combination of known ingredients with limited new scientific insight.

**Reviewer Concerns:**

The rebuttal substantively addressed practical concerns by adding a detailed compute/memory analysis, extra evaluations and a transfer experiment on BLIP2, and it also provided a lemma that offers some intuition for hierarchical fusion stability; however, the core concerns about novelty and deeper explanatory/theoretical contribution remain only partially resolved (the lemma is intuitive rather than a strong guarantee, and the novelty argument is largely positioning).

**Reviewer Scores:**

I expect Reviewer 9peL already demonstrated the likely outcome by moving from 4→6 after seeing the new analyses, while X6cr and 2gBF would plausibly increase modestly (e.g., +1) given the added attacks, cost accounting, and BLIP2 results; in contrast, the very negative reviewer hWqM (score 2, high confidence) is unlikely to substantially change because their main blockers were novelty and lack of strong theory/insight, which the rebuttal improves but does not convincingly overturn.

---

### Decision · Program_Chairs · 2026-01-26

Reject